# Probing multiphoton light-induced molecular potentials

M. Kübel 1,2,3✉, M. Spanner[1], Z. Dube[1], A. Yu. Naumov[1], S. Chelkowski[4], A. D. Bandrauk[4], M. J. J. Vrakking 5, P. B. Corkum[1], D. M. Villeneuve 1 & A. Staudte 1✉

The strong coupling between intense laser fields and valence electrons in molecules causes distortions of the potential energy hypersurfaces which determine the motion of the nuclei and influence possible reaction pathways. The coupling strength varies with the angle between the light electric field and valence orbital, and thereby adds another dimension to the effective molecular potential energy surface, leading to the emergence of light-induced conical intersections. Here, we demonstrate that multiphoton couplings can give rise to complex light-induced potential energy surfaces that govern molecular behavior. In the laser-induced dissociation of $H_2^+$, the simplest of molecules, we measure a strongly modulated angular distribution of protons which has escaped prior observation. Using two-color Floquet theory, we show that the modulations result from ultrafast dynamics on light-induced molecular potentials. These potentials are shaped by the amplitude, duration and phase of the dressing fields, allowing for manipulating the dissociation dynamics of small molecules.

[1] Joint Attosecond Science Laboratory, National Research Council and University of Ottawa, 100 Sussex Drive, Ottawa, ON K1A 0R6, Canada. [2] Department of Physics, Ludwig-Maximilians-Universität Munich, Am Coulombwall 1, D-85748 Garching, Germany. [3] Institute for Optics and Quantum Electronics, University of Jena, Max-Wien-Platz 1, D-07743 Jena, Germany. [4] Laboratoire de Chimie Théoretique, Faculté des Sciences, Université de Sherbrooke, Sherbrooke, QC J1K 2R1, Canada. [5] Max-Born-Institute, Max-Born-Straße 2A, D-12489 Berlin, Germany. ✉email: Matthias.kuebel@uni-jena.de; Andre.Staudte@nrc-cnrc.gc.ca

Potential energy surfaces describe the forces acting on the nuclei of a molecule. Within the Born–Oppenheimer approximation the motion of the nuclei along these potentials is treated independently of the electronic motion. This picture breaks down when the electronic level separation becomes comparable to the kinetic energy of the nuclei. This occurs at specific points in the molecular geometry, which are known as conical intersections and that are a hallmark of polyatomic molecules[1]. Conical intersections play an eminent role in visible and ultraviolet photochemistry[2,3], for example, in isomerization[4,5], and electron transfer processes[6]. Moreover, they are strongly implicated in the photostability of DNA by way of allowing radiation-less de-excitation[7].

The single-photon transition between two dipole-coupled electronic states can also create a conical, albeit transient, intersection. Hence, these localized features of the laser-dressed potential energy surface have been dubbed light-induced conical intersections (LICI)[8,9]. Their precise position, and the underlying dipole coupling strength are determined by the frequency and intensity of the incident light. LICIs can also be found in diatomic molecules, since the angle between the light polarization and the molecular axis adds another degree of freedom to the nuclear motion[10,11]. The angle-dependent distortion of the molecular potential energy surfaces in a linearly polarized laser field directly affects molecular dissociation[12–16] and has been predicted to cause rotational excitation[17–22]. Recently, experimental indications of LICI in $H_2^+$ have been found in angle-resolved ion spectra[23].

In ultrashort infrared laser fields, the light intensity can easily exceed the threshold for multiphoton transitions. While LICIs are a consequence of single-photon couplings and therefore the potential energy scales linearly with respect to variations of the laser field strength, multiphoton couplings lead to unique structures of their own. In the case of diatomic molecules, these structures become nonlinear point intersections of the potential energy surfaces. The one-dimensional (1D) treatment of single and multiphoton resonances has led to the prediction of light-induced potentials (LIPs)[23–29], and anomalous fragment angular distributions have been predicted in the non-perturbative regime[30]. Experimentally, however, the consequences of the angle-dependent coupling strength around nonlinear point intersections for the dissociation dynamics have so far been largely unexplored.

Here, we show in theory and experiment that LIPs featuring nonlinear light-induced point intersections can result in strong modulations of the angular ion yield. Using a two-color pump–probe scheme allows us to probe and control the nuclear dynamics, as the underlying LIPs evolve in the laser field. The experimental results are interpreted with the help of numerical solutions of the time-dependent Schrödinger equation and two-color Floquet theory. We find that, owing to both linear and nonlinear transitions, as well as rotational dynamics, the two-color laser field gives rise to remarkably complex dissociation dynamics that produce the strong modulations in the angular ion yield.

For our studies, we choose the simplest molecule, $H_2^+$, which is widely regarded as a prototype system for the interaction of molecules with light[31]. Due to the sparsity of electronic states, $H_2^+$ can often be described as a two-level system consisting of the two lowest electronic states, $1s\sigma_g$ and $2p\sigma_u$. When coupled to intense laser fields, these states give rise to intensity-dependent dissociation mechanisms known as bond softening[32] and above threshold dissociation[33]. The possibility to control them using the laser amplitude, frequency, phase, and pulse duration has been demonstrated[34–38]. In particular, the opposite parity of the $\sigma_g$ and $\sigma_u$ states can lead to electron localization[39], giving rise to charge resonance-enhanced ionization[40,41] and symmetry breaking in dissociation[42–44] under the influence of two-color laser fields[45,46] or carrier-envelope phase (CEP) stable few-cycle pulses[47–50]. Notably, the dissociation dynamics are sensitive to the rather complex shape of orthogonally polarized two-color fields[51]. Descriptive treatments for most of these phenomena are provided by dressed-state pictures, such as LIPs.

## Results

**Nonlinear point intersections.** Figure 1a shows an example of some LIP energy surfaces for $H_2^+$ in a moderately intense ($3 \times 10^{13}$ W/cm²), visible (685 nm) laser field, calculated using the procedure outlined in the "Methods" section. Shown are the laser-dressed states $\sigma_g$ and $\sigma_u - 1\hbar\omega_{VIS}$ i.e., $\sigma_u$, shifted down due to the absorption of one photon from the visible field. Along the laser polarization, i.e., $\theta = 0, \pi$, the state crossing indicated in Fig. 1b opens up and turns into an avoided crossing. This necessarily lowers the potential barrier at the avoided crossing and permits dissociation of formerly bound molecules (bond softening). Importantly, no such avoided crossing occurs when the laser polarization is perpendicular to the internuclear axis. Therefore, a LICI is formed at the internuclear distance where the laser-dressed $\sigma_g$ and $\sigma_u - 1\hbar\omega_{VIS}$ states cross.

While single-photon transitions dominate in moderately intense visible laser fields, multiphoton transitions become relevant when the wavelength is shifted into the mid-infrared[52]. For example, the three-photon transition by 2300 nm light becomes significant already at an intensity as low as $5 \times 10^{12}$ W/cm², see Supplementary Fig. 1 for details. Several crossings of potential curves that correspond to multiphoton transitions between the $\sigma_g$ and $\sigma_u$ states of $H_2^+$ at a wavelength of 2300 nm are shown in Fig. 1c. The corresponding LIP energy landscape calculated for an intensity of $3 \times 10^{13}$ W/cm² is presented in Fig. 1d. The potential energy surfaces exhibit complex structures that result from multiphoton couplings. Indicated in the figure are the curve crossings due to $n$-photon ($n = 1, 3$, and 5) transitions. Notably, the intersections for $n = 3$ and 5 are not conical (see Supplementary Fig. 2), as would be the case for all higher intersections. In order to see how these light-induced structures affect the nuclear dynamics, we first solve the two-dimensional (2D) time-dependent Schrödinger equation (2D TDSE, see "Methods" section for details) for $H_2^+$ under the influence of a moderately intense mid-IR laser pulse. The calculated proton momentum distributions presented in Fig. 1e show distinct features in the proton angular distribution that can be associated with the $n$-photon couplings. While these features are absent in the single-photon coupling regime at low intensity, significantly higher intensities produce very convoluted dissociation patterns that involve high-order couplings, but would likely defy experimental resolution, see Supplementary Fig. 3.

**Structured proton angular distribution.** In order to experimentally probe the light-induced molecular potentials depicted in Fig. 1d, e, describing the situation where the mid-IR field induces multiphoton dynamics in the dissociation process, but does not cause ionization, we implement the two-pulse scheme depicted in Fig. 2a. First, an intense, few-cycle visible laser pulse ionizes neutral $H_2$, producing a bound coherent wave packet in $H_2^+$ with a nearly isotropic alignment distribution, with respect to the laser polarization[53]. Second, a moderately intense mid-IR pulse creates the LIPs on which dissociation occurs. The LIPs are probed by recording the momentum distribution of protons resulting from the dissociating part of the molecular wave packet. The molecular ions dissociate along their initial alignment direction, unless rotational dynamics occur, as predicted, e.g., in refs. [17,18,21,22].

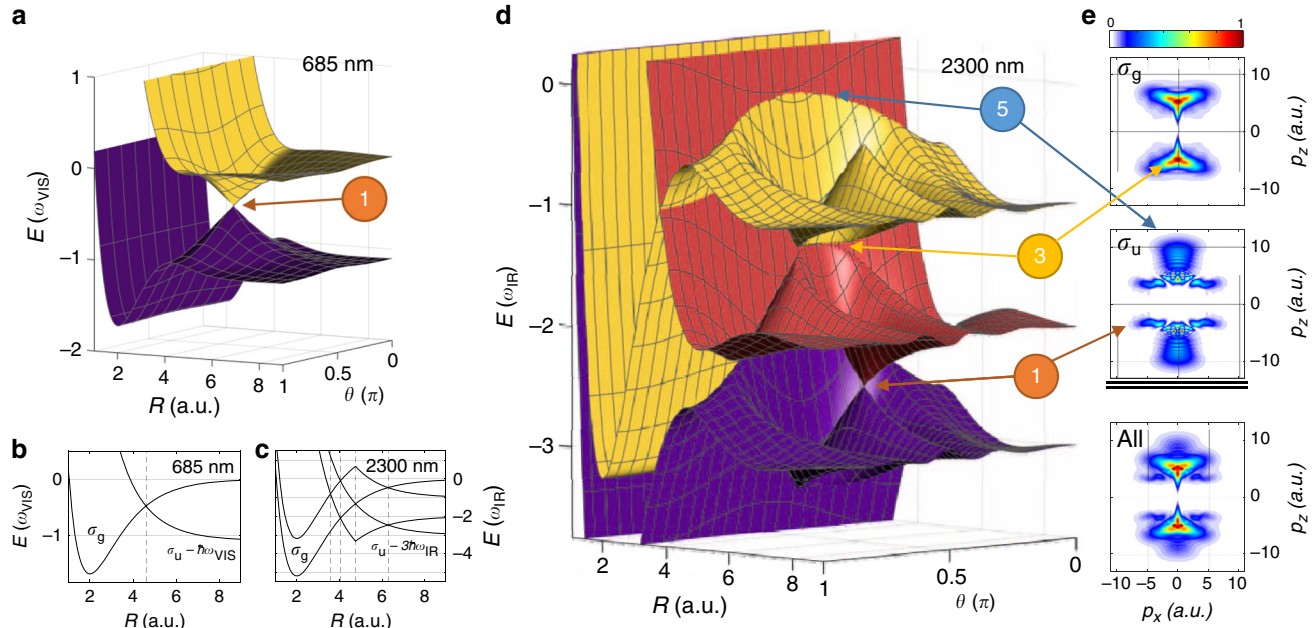

**Fig. 1 Diabatic single- and multiphoton light-induced molecular potentials. a** Angle-dependent potential energy surfaces of $H_2^+$ dressed by a linearly polarized, moderately intense, visible (685 nm, $3 \times 10^{13}$ W/cm²) laser field. The nuclear potential energy $E$ is plotted as a function of the internuclear distance $R$, and the angle $\theta$ between the molecular axis and the laser polarization. The LICI at the one-photon crossing is marked by a 1. The potential energy curve shown in **b** corresponds to a lineout along the internuclear distance $R$ through the LICI. The dotted line marks the position of the one-photon resonance between the $\sigma_g$ and $\sigma_u$ states. The potential energy diagram in **c**, shows several multiphoton crossings between the $\sigma_g$ (dissociation limits of (..., 0, −2, −4,...) $\omega_{IR}$) and $\sigma_u$ (dissociation limits of (..., 1, −1, −3, ...) $\omega_{IR}$) states in a 2300 nm dressing field. Two of these states are labeled, and the location of multiphoton transitions of order 7, 5, 3, and 1 (from left to right) are indicated by dashed lines. **d** Same as **a** for a mid-IR dressing field (2300 nm, $3 \times 10^{13}$ W/cm²). Structures attributed to 1, 3, and 5 photon couplings are marked. **e** Calculated proton momentum distributions produced by a 2300 nm, $3 \times 10^{13}$ W/cm² laser field polarized along the z-axis. The color scale indicates the proton yield. Contributions from dissociation on the $\sigma_g$ and $\sigma_u$ surfaces are shown separately, and structures attributed to the 1, 3, and 5 photon couplings are indicated.

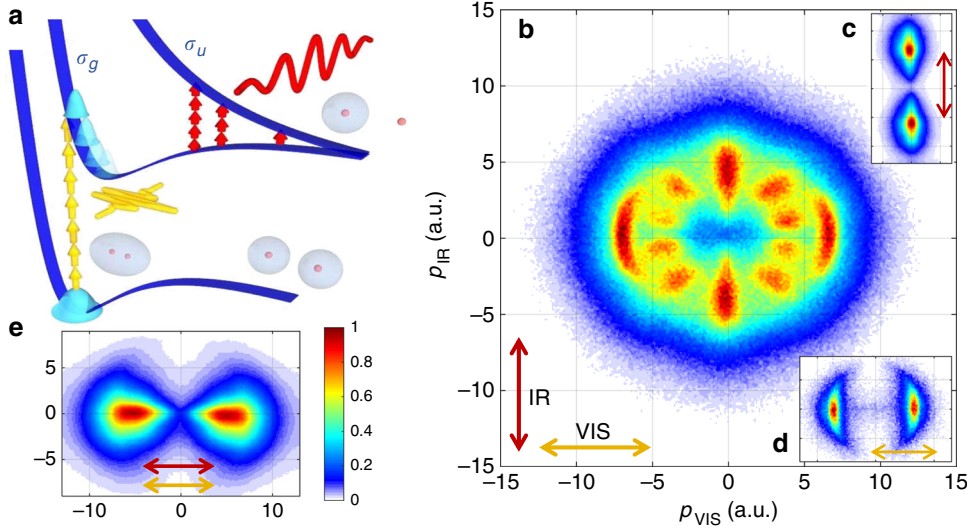

**Fig. 2 Experimental signatures of LIPs. a** A few-cycle visible pulse (yellow) ionizes $H_2$ and prepares a wave packet on the $\sigma_g$ state of the molecular cation. An additional mid-infrared control pulse (red) couples the $\sigma_g$ and $\sigma_u$ states by 1, 3, or 5 photons, creating LIPs, on which $H_2^+$ dissociates into an H atom and a proton. **b** The measured proton momentum distribution in the recoil frame for perpendicularly polarized two-color fields (730 nm, 5 fs, $2 \times 10^{14}$ W/cm², and 2300 nm, 45 fs, $3 \times 10^{13}$ W/cm²) probes the LIPs. It strongly contrasts results obtained for **c** only mid-IR pulses (2000 nm, 65 fs, $1 \times 10^{14}$ W/cm²) or **d** only visible (730 nm, 5 fs, $2 \times 10^{14}$ W/cm²) pulses. The angular structure is moreover absent in two-color experiments carried out with parallel polarization (**e**). The data has been integrated over the direction perpendicular to the polarization plane. The arrows indicate the polarization axes of the visible (orange) and mid-IR (red) laser pulses. The color scale represents the measured proton yield normalized to the maximum value in each plot.

The intent of the two-pulse scheme is to decouple the production of the molecular wave packet from the field that generates the LIPs. This allows for probing the LIPs at selected times within the mid-IR pulse by scanning the time delay between the laser pulses. Moreover, the use of a shorter wavelength pulse for ionization allows us to reduce the focal volume averaging in the long-wavelength field, which often washes out subtle features in strong-field experiments (e.g., ref. [54]). Finally, choosing a perpendicular relative polarization of the visible and mid-IR pulses is expected to avoid overlap between the signal of interest produced by the mid-IR pulse, and any protons produced by the visible pulse alone. The experimental setup is described in the "Methods" section "Experiment".

Figure 2b shows experimental results obtained with the cross-polarized few-cycle visible and mid-IR pulses. The three-dimensional (3D) momentum distributions of protons and electrons were measured in coincidence, using Cold Target Recoil Ion Momentum Spectroscopy (COLTRIMS). The coincidence measurement allows us to present results in the recoil frame, where the impact of the electron recoil has been largely removed from the measured ion momentum distribution. The results exhibit a striking angular structure that is blurred if the recoil momentum is not accounted for (see Supplementary Fig. 6). The angular structure consists of on-axis features along either of the polarization axes, and additional spots at intermediate angles. Drawing a comparison to results obtained with only mid-IR (Fig. 2c) or visible (Fig. 2d) pulses, suggests that the on-axis features arise from bond softening by either pulse alone. Note, that the signal along the mid-IR polarization in the two-color experiment does not arise from dissociative ionization of neutral $H_2$ by the mid-IR pulse on its own, as no notable ionization of neutral $H_2$ is obtained at the intensity of $3 \times 10^{13}$ W/cm². Hence, the comparative mid-IR only data (Fig. 2c) is presented for a higher intensity of $1 \times 10^{14}$ W/cm². The additional spots in the two-color data are tentatively attributed to dynamics caused by the light-induced structures in the molecular potential energy landscape; cf. Fig. 1d.

Surprisingly, the experimental results presented in Fig. 2b, exhibit a much more pronounced angular structure than the TDSE results for the mid-IR field alone, presented in Fig. 1e. Moreover, the angular structure survives averaging over kinetic energy, in contrast to the weaker modulations in previous work on LICI (ref. [23]). A hint on the origin of the additional angular structure in the present experiment comes from measurements carried out with a parallel polarization of the visible and mid-IR pulses. The proton momentum distribution obtained with parallel polarization is shown in Fig. 2e. It resembles the results obtained with mid-IR pulses only and does not exhibit significant angular structure. Although the intention of our scheme was to decouple the effects of the mid-IR and visible pulses, the striking dependence of the angular structure on the relative polarization of the two laser fields implies that the visible field contributes to the formation of the additional spots. It will thus be considered in the following analysis of our results.

**Numerical results**. In a first step to understand the dynamics producing the structured proton angular distribution observed in cross-polarized fields, we solve the 2D TDSE for $H_2^+$, taking both laser pulses into account. Due to the observed importance of the visible field in shaping the experimental results, we also consider a weak pulse pedestal at 5% of the peak intensity and 45 fs duration (full-width at half-maximum of the intensity envelope) for the visible few-cycle pulse. These values are consistent with field-resolved measurements of few-cycle pulses[55].

The initial alignment of the molecular axis with respect to the laser polarization is assumed to be istropic.

It has been recognized in the literature that angular modulations in the proton spectra can arise from rotational dynamics in the vicinity of the LICI (refs. [18,21–23]); more specifically, simulations that include rotational motion in the dissociation dynamics show angular modulations that are absent when the rotational degrees of freedom are frozen. These modulations can be connected to rotational scattering of the dissociating wave packet from the LICI (refs. [18,21–23]). A first candidate for the physical mechanism underlying the appearance of a structured proton angular distribution is therefore the formation of a high-order rotational wave packet in the dissociating molecular cation. In order to test the role of rotational dynamics, we perform a first set of calculations where rotational transitions are artificially switched off and present the results in Fig. 3a. Evidently, pronounced modulations in the angular distribution are obtained, even without the inclusion of wave packet rotation. This suggests that rotational dynamics are not the primary physical mechanism underlying the angular modulation in the proton momentum distribution. Therefore, it will be important to identify how angular modulations arise already within a 1D treatment.

The second set of calculations (Fig. 3b) takes rotational dynamics into account. The differences between Fig. 3a, b show the impact of rotations in certain parts of the momentum distributions. The black arrows highlight pronounced differences between the results of the two calculations at angles $\theta = 0°$ and $\theta = 20°$. These differences illustrate the role of rotational alignment in shaping the final momentum distribution. On the basis of the comparison between Fig. 3a, b, we conclude that rotational dynamics play a significant but secondary role in defining the final momentum distribution; in contrast with the previously considered pure LICI case, the addition of rotations is not the sole cause of the angular structures in our experiment. A direct comparison of the calculated and measured angular distributions is given by Fig. 3d. Indeed, the strong modulations observed in the experimental data are only obtained in the simulations that take rotations into account. However, the modulation depth in the experimental data is smaller than in the simulations with rotations, which is ascribed to the reduced dimensionality of the simulations.

In order to identify the essential mechanism creating the angular structure in the absence of rotations, we employ two-color Floquet theory (see "Methods" section). We calculate the angle-dependent field-dressed states of $H_2^+$ using a two-color laser field, $\mathbf{F}(t) = \sqrt{I_{VIS}} \cos(\omega_{VIS} t + \varphi_{VIS})\hat{\mathbf{x}} + \sqrt{I_{IR}} \cos(\omega_{IR} t + \varphi_{IR})\hat{\mathbf{z}}$ ($\hat{\mathbf{x}}$ and $\hat{\mathbf{z}}$ being the unit vectors along $x$- and $z$-directions, respectively).

The field consists of a moderately intense mid-infrared field ($\lambda_{IR} = 2280$ nm, $I_{IR} = 3 \times 10^{13}$ W/cm²) and a weak visible field ($I_{VIS} = 1 \times 10^{13}$ W/cm²), corresponding to the pulse pedestal used in the TDSE calculations. We take $\lambda_{VIS} = \lambda_{IR}/3$ to ensure the periodicity of the laser fields required by Floquet theory. The resulting LIP energy landscape depends on the relative optical phase, $\Delta\varphi = \varphi_{VIS} - \varphi_{IR}$. Exemplarily, we present the LIP energy landscape for $\Delta\varphi = 0$ in Fig. 3d. Both experimental and numerical results, presented in Figs. 2 and 3, respectively, are integrated over $\Delta\varphi$.

A detailed analysis and discussion of the results from Floquet theory is presented in Supplementary Note 2. In brief, we find that the Floquet states represent a conclusive basis for understanding the emergence of angular structure in the proton momentum distribution, even without rotational dynamics taken into account. In the absence of rotations, the angular structure arises through a process we shall call angle-dependent channel switching, as different orders of multiphoton couplings dominate

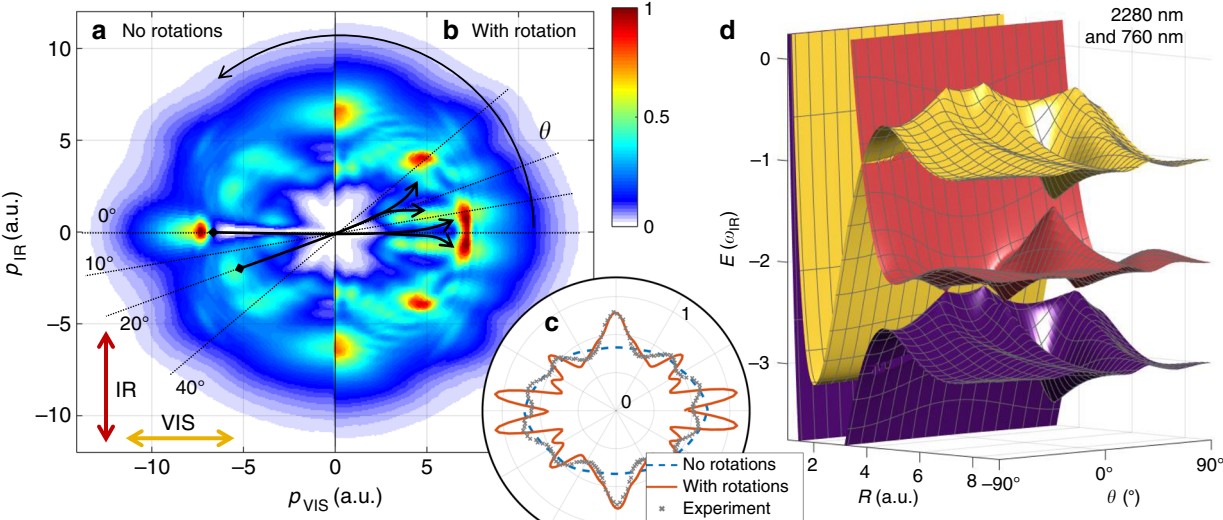

**Fig. 3 Numerical results for two-color bond softening of $H_2^+$.** Proton momentum distributions obtained from solving the time-dependent Schrödinger equation **a** without and **b** with rotational dynamics taken into account. The colorbar represents the proton yield per momentum bin. The black arrows indicate population transfer through rotational dynamics. The dotted black lines at various angles are drawn as a guide to the eye. **c** The calculated proton angular distributions are compared to the measured data presented in Fig. 2b. Each angular distribution is integrated over kinetic energy and normalized to its integral. The error bars (s.d.) for the experimental data are of the same size as the symbols. The calculations have been averaged over a delay range corresponding to one mid-IR cycle around the temporal overlap between the few-cycle visible pulse with the center of the mid-IR pulse. **d** The LIP energy landscape obtained from Floquet theory for a two-color field with relative phase $\Delta\varphi = 0$. The nuclear potential energy is plotted as a function of the internuclear distance $R$, and the angle $\theta$ between the molecular axis and the polarization of the visible field.

at different alignment angles of the molecular axis with respect to the laser polarization. The following picture can be invoked.

As the alignment angle of the molecular axis in the polarization plane (see Fig. 3), $\theta$, is increased, the field components parallel to the molecular axis vary as $F_{VIS} \propto \cos(\theta)$, and $F_{IR} \propto \sin(\theta)$. This leads to a pronounced angle dependence of the field-dressed potential energy curves (see Fig. 3c), where several Floquet state crossings open up and close again as $\theta$ is varied. Specifically, at $\theta = 0°$, i.e., for alignment of the molecular axis perpendicular to the mid-IR polarization, the effect of the mid-IR field is insignificant, and dissociation proceeds as in the single-color case (seen in Fig. 2b). Specifically, the wave packet dissociates on the purple surface in Fig. 3d. As $\theta$ is increased, a new dissociation channel due to one-photon coupling by the mid-IR field opens up. The new channel competes with the original one, which moves population to the pronounced feature at $\theta = 10°$, making the on-axis feature much narrower than in the single-color case. This new dissociation channel corresponds to dissociation on the red surface in Fig. 3d. As $\theta$ is further increased, the width of the avoided crossing reaches $2\omega_{IR}$, which closes the dissociation channel and gives rise to a LICI at $\theta \approx 30°$, clearly visible in Fig. 3d.

Notably, the computational results obtained without (Fig. 3a) and with (Fig. 3b) rotations strongly differ at $\theta \approx 20°$. We attribute this to the presence of the aforementioned LICI that promotes strong rotational dynamics, as the nuclear wave packet propagates around the cone in the LIP landscape. In a similar manner, the splitting of the narrow feature at $0°$ in Fig. 3a into the double peak structure in Fig. 3b is attributed to the point intersections at $\theta = 0°$.

**Delay dependence**. Scanning the time delay between the visible and mid-IR pulses in our experiment allows us to probe the variations in the LIPs throughout the mid-IR pulse. The time delay controls the time of ionization with respect to the mid-IR pulse, and thereby determines the (i) strength, (ii) duration, and

(iii) phase of the mid-IR field at the time it interacts with the molecular ion. In Fig. 4, we analyze the fragment momentum distribution for overlap of the ionizing visible pulse with the rising edge, the maximum and the falling edge of the mid-IR laser pulse. Each of the presented spectra are integrated over two mid-IR cycles, and therefore not expected to be sensitive to the mid-IR phase.

Figure 4a shows the vector potential of the mid-IR pulse used in our experiment, as measured with the STIER technique[56] (see Supplementary Fig. 5). Selected recoil-frame proton momentum distributions are presented in Fig. 4b–d. The delay-dependent results probe the evolution of the LIP energy landscape throughout the mid-IR pulse. This is evidenced by the changes in the recorded dissociation patterns, as the delay between visible and mid-IR pulses is varied. For example, the feature at $\theta = 90°$, i.e., along the mid-IR polarization axis, peaks around the center of the pulse, Fig. 4c, where it represents the strongest contribution to the proton momentum distribution. When the ionization occurs on the falling edge of the mid-IR pulse (Fig. 4d), the $90°$ feature is absent. On the basis of the computational results presented in Fig. 1 and the two-color Floquet states shown in Fig. 3d, we attribute this peak to a five-photon coupling induced by the mid-IR pulse. The nonlinearity of this process explains why this feature is particularly visible near the maximum of the mid-IR pulse and decays rapidly on the falling edge of the pulse. Notably, the maximum yield of protons emitted at $90°$ is obtained, when the visible pulse precedes the peak of the mid-IR pulse by $(8.3 \pm 0.5)$ fs, in reasonable agreement with the 7.3 fs vibrational half-period of $H_2^+$ (ref. [57]). For earlier delays, when ionization occurs on the rising edge of the mid-IR pulse (Fig. 4b), the weaker signal at $90°$ indicates that dissociation occurs before the molecular ion interacts with the center of the mid-IR pulse. Similar observations are made for the feature at intermediate angles (around $\theta \approx 40°$ in Fig. 3a). In addition, its angular position also varies from $\theta \approx 30°$ in Fig. 4b toward $\theta \approx 40°$ in Fig. 4c.

Contrary to the nonlinear features, the feature at $\theta = 10°$, i.e., close to the visible polarization axis, exhibits little delay

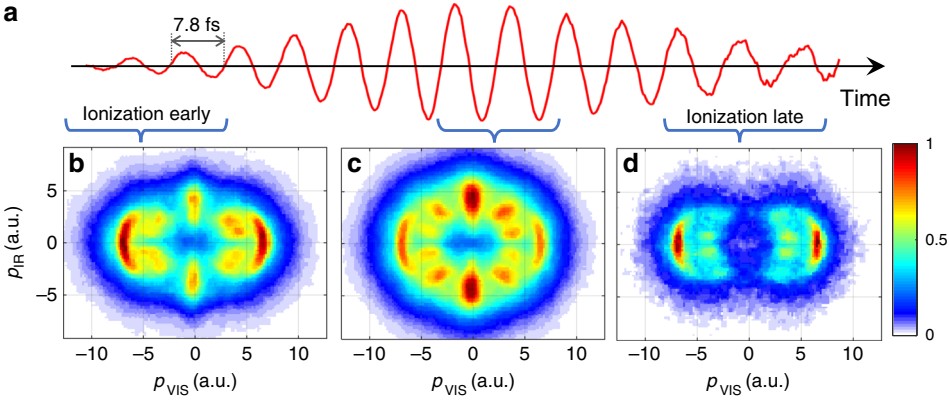

**Fig. 4 Tracking the evolution of light-induced molecular potentials in $H_2^+$ throughout a mid-IR laser pulse. a** Vector potential of the mid-IR dressing field measured using the STIER technique. **b–d** Measured $H^+$ momentum distributions in the polarization plane for different ionization times within the mid-IR pulse. The signal has been integrated over the delay ranges indicated by the brackets. The colorbar indicates the proton yield normalized to the maximum in each plot.

dependence. As discussed above, this feature can be understood as a consequence of the single-photon couplings by both, the visible and the mid-IR fields. The absence of nonlinearity in this process explains the insensitivity of the 10° feature to the mid-IR intensity. Calculated proton momentum distributions for different delay values, which are consistent with these conclusions, are presented in Supplementary Fig. 7.

## Discussion

In summary, we have demonstrated a powerful approach for probing light-induced molecular potentials. We observed strongly modulated proton angular distributions in experiments were $H_2^+$ ions, produced by a linearly polarized, few-cycle, visible laser pulse, are dissociated by a cross-polarized mid-IR laser field. We have shown that the modulations can be understood as signatures of complex LIP energy landscapes that are shaped by both single-photon and multiphoton transitions in a cross-polarized two-color laser field. Specifically, the modulations arise from a combination of two effects: First, angle-dependent channel switching, i.e., different dissociation pathways open and close as a function of alignment angle; second, rotational motion around light-induced point intersections, such as LICIs, shape the modulated angular ion yield. The LIP picture predicts where angle-dependent channel switching takes place, and where prominent light-induced point intersections are present.

Probing the LIPs resulting from the mid-IR dressing field on its own may be improved by using a shorter pulse for preparation of the bound wave packet, such as a few-cycle UV or attosecond pulse. Previous experiments along these lines (e.g., refs. [44,58]) were conducted in the single-photon dressing regime and did not study the influence of the LIP surfaces on the angular dependence of dissociation.

Our approach allows us to follow the variation of the LIPs throughout the dressing laser field. On the timescale of the mid-IR pulse envelope, we observe the opening and closing of dissociation pathways as the dressing field strength changes. On shorter time scales, the propagation of the dissociating wave packet will become accessible with sub-femtosecond time resolution by monitoring the electron localization on either fragment. More generally, we have shown how complex LIP energy landscapes determine the outcome of molecular dissociation, using $H_2$ as an example. Our approach will allow for elucidating the reaction dynamics of more complex molecules in the presence of LICIs and higher-order point intersections.

## Methods

**Experiment.** The employed experimental technique is a variant of ref. [56]. The output of a commercial Ti:Sa chirped pulse amplification (CPA) laser (Coherent Elite, 10 kHz, 2 mJ) is split into two parts. The stronger part (85%) is used to pump an optical parametric amplifier, in order to obtain CEP stable idler pulses at 2.3 μm. The second part of the CPA output is focused into an argon-filled hollow core fiber to obtain broadband laser pulses, which are subsequently compressed to a pulse duration of ≈5 fs. The laser pulses are recombined using a polished Si mirror (thickness 2.2 mm) at 60° angle of incidence.

After recombination, the pulses are focused in the center of a COLTRIMS[59], where they intersect an ultrasonic jet of pre-cooled ($T = 60$ K) of neutral $H_2$. The intensity of the mid-IR pulse is weak enough to not cause any notable ionization by itself. Because ions are only produced in the small focal volume of the visible pulse ($1/e^2$ width ($7 \pm 2$) μm), focal volume averaging within the larger focal volume (($30 \pm 10$) μm) of the mid-IR pulse is essentially avoided. In the COLTRIMS, the 3D momenta of ions and electrons generated in the laser focus are measured in coincidence, which provides access to the recoil-frame ion momentum that arises solely from the nuclear dynamics on the LIPs. See Supplementary Fig. 6 for a comparison of laboratory-frame and recoil-frame measurements. The measurement of the delay dependence of the electron momentum distribution yields the instantaneous mid-IR vector potential at each delay value, as shown in Supplementary Fig. 5.

**Time-dependent Schrödinger equation.** For the dynamics in the $H_2^+$ cation, we solve a 2D (one angle and one bond length) Schrödinger equation that includes dipole coupling between the two relevant electronic states $^2\Sigma_g^+$ (also referred to as $\sigma_g$) and $^2\Sigma_u^+$ ($\sigma_u$)

$$i\frac{\partial}{\partial t}\begin{bmatrix} \Psi_g(R) \\ \Psi_u(R) \end{bmatrix} = -\frac{1}{2\mu}\left(\frac{\partial^2}{\partial R^2} + \frac{1}{R^2}\frac{\partial^2}{\partial \theta^2}\right)\begin{bmatrix} \Psi_g(R) \\ \Psi_u(R) \end{bmatrix} + \begin{bmatrix} V_g(R) & -F(t)\cdot d(R) \\ -F(t)\cdot d(R) & V_u(R) \end{bmatrix}\begin{bmatrix} \Psi_g(R) \\ \Psi_u(R) \end{bmatrix}, \quad (1)$$

where $R = (R,\theta)$ are the bond length, and angle between the laser field and the molecular axis. $F(t)$ is the electric field of the laser that couples the two electronic states. The form of the electronic potential energy curves $V_g$ and $V_u$, as well as the transition dipole d, are taken from Bunkin and Tugov[60]. Equation (1) was solved numerically using the Fourier split-operator method.

In our experiment, the $H_2^+$ system is created starting from the $H_2$ neutral through strong-field ionization. The initial state of the wave function of the ionic simulations assumes a vertical transition from the ground electronic ($^1\Sigma_g$), and ground vibrational state of the $H_2$ neutral to the ground electronic state of the ion. The ground vibrational state on the $^1\Sigma_g$ of the neutral is modeled as Morse oscillator state, using Morse parameters derived from Herzberg[61]. The rotational degree of freedom was initialized to a thermal rotational distribution, with temperature chosen to be low enough such that only the rotational ground state is populated. The initial distribution of the molecular axis with respect to the laser polarization is isotropic, closely reflecting the experimental conditions.

The laser field used in the calculations presented in Fig. 3a, b can be expressed as

$$F(t) = F_{IR}(t + \Delta t)\hat{z} + (F_{VIS}(t) + F_{ped}(t))\hat{x}, \quad (2)$$

where $\Delta t$ is the time delay between visible and mid-IR pulses and each field, $F_A(t)$, is

given by an expression (using atomic units),

$$F_A(t) = \sqrt{I_A} \exp\left(-2\ln 2\left(\frac{t}{\tau_A}\right)^2\right)\cos(\omega_A t + \varphi_A), \qquad (3)$$

where $\varphi_A$ is the CEP of each pulse.

The laser field consists of a mid-IR pulse ($\lambda_{IR} = 2300$ nm, $\tau_{IR} = 45$ fs, $I_{IR} = 30$ TW/cm$^2$), an ionizing few-cycle visible pulse ($\lambda_{VIS} = 730$ nm, $\tau_{VIS} = 5$ fs, $I_{VIS} = 300$ TW/cm$^2$), and a visible pulse pedestal ($\lambda_{ped} = 730$ nm, $\tau_{ped} = 45$ fs, $I_{ped} = 10$ TW/cm$^2$)). The calculations are started at $t = 0$, i.e., in the center of the visible pulse and performed for various values of $\Delta t$ and, for each $\Delta t$, $\varphi_{VIS} = \varphi_{ped} = n\pi$ ($n = 0, 1$) and $\varphi_{IR} \equiv 0$.

**Floquet states.** For each molecular alignment angle $\theta$, the Floquet states[62–64] are calculated for a field

$$F(t, \theta) = \sqrt{I_{IR}}\cos(\omega t)\sin\theta + \sqrt{I_{ped}}\cos(3\omega t + \phi)\cos\theta, \qquad (4)$$

where $\phi$ is the relative phase of the two fields. The potential energy landscape presented in Fig. 3b is for the relative phase $\phi = 0$. Here, the frequency of the visible pulse is approximated as $\omega_{VIS} = 3\,\omega_{IR}$ to obtain the required periodicity.

At each point along $R$, the Floquet states were constructed as follows. First, the one-period propagator $U(t, t + T; R)$, where $T = 2\pi/\omega$ is the period of the 2280 nm laser field, was constructed numerically using

$$U(t, t+T; R) = e^{-iH_e(R,t_{N-1})\Delta_t}e^{-iH_e(R,t_{N-2})\Delta_t} \cdots e^{-iH_e(R,t_1)\Delta_t}e^{-iH_e(R,t_0)\Delta_t}, \qquad (5)$$

where the time interval $\tau$ has been split into $N = 1024$ time steps of duration $\Delta_t = T/N$ with the intermediate times given by $t_n = t + n\Delta_t$, and the purely electronic Hamiltonian $H_e(t)$ is given by

$$H_e = \begin{bmatrix} V_g(R) & -F(t)\cdot \mathbf{d}(R) \\ -F(t)\cdot \mathbf{d}(R) & V_u(R) \end{bmatrix}. \qquad (6)$$

The Floquet states $|F_\alpha(R, t)\rangle$ are the eigenstates of $U(t, t + T; R)$,

$$U(t, t+T; R)|S_\alpha(R, t)\rangle = e^{-i\varepsilon_\alpha(R)t}|S_\alpha(R, t)\rangle, \qquad (7)$$

where the $\varepsilon_\alpha(R)$ are the quasi-energies of the Floquet states $|S_\alpha(R, t)\rangle$. The Floquet states and quasi-energies are found directly by diagonalizing the $2 \times 2$ $U(t, t + T; R)$ matrix for each $R$. The Floquet states $|S_\alpha(R, t)\rangle$ are periodic with the period of the laser field, and exhibit a sub-cycle time dependence whenever multiphoton couplings are active. Consequently, the associated potential energy surfaces will also, in general, exhibit a sub-cycle time dependence. The sub-cycle time dependence can be expanded as a Fourier series to yield a set of time-independent potentials that characterize the system

$$e^{-i\varepsilon_\alpha(R)t}|S_\alpha(R, t)\rangle = e^{-i\varepsilon_\alpha(R)t}\sum_{n=-\infty}^{\infty}|s_\alpha^n(R)\rangle e^{-in\omega t} = \sum_{n=-\infty}^{\infty}|s_\alpha^n(R)\rangle e^{-i(\varepsilon_\alpha(R)+n\omega)t}. \qquad (8)$$

The ladder of Floquet states is formed by the energies of the Fourier expansion, where the $\varepsilon_\alpha(R)$ get repeated and shifted by $n\omega$, forming an infinite ladder of time-independent potentials. These quasi-energies ($\varepsilon_\alpha(R) + n\omega$) are what is referred to as LIPs in the main text.

## Data availability
The data that support the findings of this study are available from the corresponding author upon reasonable request.

## Code availability
The computer codes used for TDSE simulations and Floquet calculations are available from the corresponding author upon reasonable request.

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

## Acknowledgements

The authors thank D. Crane, R. Kroeker, and B. Avery for technical assistance. We acknowledge fruitful discussions with F. Bouakline, M. Richter, A. M. Sayler, G. G. Paulus, M. F. Kling, and B. Bergues. This project has received funding from the EU's Horizon2020 research and innovation programme under the Marie Sklodowska-Curie Grant Agreement No. 657544. Financial support from the National Science and Engineering Research Council Discovery Grant No. 419092-2013-RGPIN, and from the U.S. Air Force Office of Scientific Research (Grant No. FA9550-16-1-0109) is gratefully acknowledged.

## Author contributions

M.K., Z.D., and A.S. conceived and conducted the experiment, and analyzed the results. M.S., M.K., S.C., M.J.J.V., and D.M.V. performed simulations, and interpreted the data with P.B.C. and A.S. All authors discussed the results and contributed to the final manuscript.

## Competing interests

The authors declare no competing interests.
