## [Peer Review File · Nature Communications]

Reviewers' comments:

Reviewer #1 (Remarks to the Author):

Report on:

“Probing multiphoton light-induced molecular potentials”

M. Kübel, M. Spanner, Z. Dube, A.Yu. Naumov, S. Chelkowski, A.D. Bandrauk, M.J.J. Vrakking, P.B. Corkum, D.M. Villeneuve, and A. Staudte

This paper reports on an experimental and theoretical investigation for probing light-induced molecular potentials by using two-color few-cycle visible and mid-IR pulses. Although the intention of their two-pulse scheme was to decouple the effect of the ionization VIS and mid-IR (creates the LIPs) pulses, but the strong dependence of the angular distribution of the photofragments on the relative polarization of the two laser fields implies that the visible pulse still contributes to the formation of the final structure.

The main conclusion of this work is that the modulations of the proton angular distributions are the fingerprint of complex LIP landscapes which are formed by both single and multi-photon transitions. Namely, these are joint effects of two different phenomena: i) dissociation pathways for a given mid-IR laser intensity open and close as a function of alignment angle (multi-photon effect), ii) rotational motion around the LICIs.

2 dimensional quantum-dynamical numerical simulations support the interpretation of the experimental results. Two-color Floquet theory helps to reveal the basic mechanism creating the additional angular structure (structure in the absence of rotation) of the photofragments.

The paper is rich and overall well written. The figures are clear, the analysis of the results is thorough and the conclusions of the study are interesting and important. I recommend publication in Nature Communication after the authors have considered the revisions suggested below.

i) In Fig. 3 (page 12) the proton momentum distributions obtained by solving the TDSE without and with rotation are shown. Here the black arrows indicate the population transfer through rotational dynamics. I would suggest to give some numbers to announce the magnitude of the total dissociation products. It would be very informative. It is true that there are modulations in the angular distribution even without including the rotation, but in spite of this, it seems that the dissociation products can be significantly different in the non-rotational and the 2D models.

ii) The expressions (on page 17) are confusing: the definition of $F(t)$ in Eq. 1 is missing; following there are two different formulas with the same $E(t)$ on left side.

iii) The authors may wish to cite the following references [G. J. Halász et al, J. Phys. Chem. A 117, 8528 (2013); G. J. Halász et al, Phys. Rev. A 88, 043413 (2013)] using a similar theoretical framework and where the initial vibrational wavepacket is defined as the Franck–Condon distribution of the vibrational states of the ion.

iv) Small typos should be corrected.

Reviewer #2 (Remarks to the Author):

This manuscript is about a combined theory/experiment work on the creation and probing of multiphoton (non-linear) light induced potentials, which are revealed by the observation (and simulation) of modulated angular distributions of protons arising from the photodissociation of the

simplest molecular system, the hydrogen ion molecule, H₂⁺. The experiment employs state-of-the-art photoion photoelectron coincidence (COLTRIMS) measurements in pump-probe experiments carried out with carrier-envelope-phase stabilized IR pulses at 2.3 μm and ultrashort 5 fs IR pulses. The theoretical treatments are also state-of-the-art and support and help to understand the experimental observations. The paper is well written and presented and the conclusions are supported in general by the experimental results and numerical simulations.

I do not have major objections for the paper to be accepted for publication in Nature Comm.; however, the authors should present more convincing arguments: What is the real message presented in the paper? Is it that from the observation of fragment angular distributions in coincidence at different time delays they show that the photodissociation process is more complex than expected for such a simple system? I do not see that this allows us to “visualize” the LIP. On the contrary, this indicates that there are many non-adiabatic processes in between LIPs. In this sense, the LIP concept is not so useful. Comments should be added in the manuscript.

In addition, more information should be included considering that they work with angular momentum J=0 (as I understood) which implies no initial rotation of the molecule, but there is an isotropic distribution with respect to the laser polarization.

Technically, I do not understand how the LIPs are calculated from Floquet according to the equations in the section of Floquet states (by the way, equations should be numbered!). The authors say that $\Delta t = \tau/N$ but there is no value given for N. It seems that this equation goes beyond the rotating wave approximation (RWA), but then only allows us to obtain 2 LIPs. How the authors can get all LIPs that cross at different hw_{IR} values? (for instance, those that are shown in Fig. 1c with $\sigma_u - 3hw_{IR}$, etc.). There is something missing here.

In conclusion, I can recommend publication of the manuscript but the authors should address the above comments in a revised version of the paper.

Reviewer #3 (Remarks to the Author):

The authors investigate the light induced conical intersection in the dissociative ionization of H₂ molecules by observing a modulated angular distribution of the protons. A pump-probe scheme is used, where the pump ionization created H₂⁺ is dissociated by the probe pulse. The modulations in the angular distributions of the strong-field photo-dissociation of H₂⁺ has been measured several years ago, for instance, prl 116, 143004 (2016), where a 30 fs pulse at 795 nm was used to dissociate a H₂⁺ beam. The produced H and H⁺ fragments were measured in coincidence using a time- and position-sensitive detector, similar to the present work. The authors claimed that they investigated the non-linear light-induced point intersections in dissociative ionization of H₂, resulting in strong modulations of the angular ion yield. The observation of the “non-linear light-induced point intersections” is indeed an addition to previous works, e.g. prl 116, 143004 (2016), but it is not fundamentally “new” and don’t stand for a significant advancement of this field by simply adding more photons. Based on the work of “single-photon” induced conical intersection, the multi-photon case is expected straightforwardly in a similar picture. For the very limited innovation, I cannot recommend the publication of the present paper in the highly impact journal of NC.

The present paper may be suitable to be published in a more specialized journal, e.g. physical reviews. Here I have several comments that the authors may need to properly address if they transfer to other journals.

-In prl 116, 143004 (2016), it was noticed that the modulated angular distribution depends on the vibrational levels of the initially populated H₂⁺, which will be smeared out after an integration over

various vibrational levels. Is this the case in the present experiment? Do the authors resolve the vibrational levels of the pump ionization created H₂⁺? I guess this effect may be more serious for the multiphoton induced conical intersection than the sing-photon process.

-I am a little confused by the "role" of the rotational dynamics. The authors claim that the point section or LICI promotes the rotational dynamics. However, they also claim that the rotational dynamics plays "secondary role in defining the final momentum distribution". The agreement between of the simulation results shown in figure 3(a) is better when the rotational dynamics is switched off. The simulated splitting around +/-10 degree is not observed in the experiments shown in figure 2.

-For the results presented in figure 3, is the delay set to be zero between the few-cycle visible pulse and the IR pulse? A simulation of the delay dependent angular distribution of the protons, similar to figure 4, will be helpful for one to understand the experimental observations. As shown in figure 4, the angular distribution of the protons strongly depends on the time delay between the pump and probe pulses. For dissociation of the H₂⁺ created by the pump pulse, the LICI induced by the probe is expected to be more significant when the pump pulse coming earlier, and should also be clearly observed even there is no temporal overlap between the pump and probe, i.e. the pump is much advanced than the probe. This will be similar to previous observations in prl 116, 143004 (2016) where a H₂⁺ beam is used. The authors need to confirm this point in their experiments. This also relates to the physical mechanism of their observations, and will be helpful to exclude other effects which require the temporal overlap of the pump and probe pulses.

-A pump pulse of 5 fs was used in the experiment to ionize H₂. For such a short laser pulse, I am wondering the probability of the dissociation of the self-ionization created H₂⁺ within the same pulse. Let's assume that the ionization probability is maximal around the peak of the 5 fs pulse, and the molecular ion will not have enough time to stretch to the critical internuclear distance for the one-photon dipole transition in the falling edge of the 5 fs pulse. More explanation is required here for the observation of pronounced signal along the polarization of the pump pulse. Is it because of the existence of the pedestal of the few-cycle pulse? If this is the case, an experimental measurement of this pedestal is necessary for its important role for the experimental results.

- The writing of the paper is not carefully checked with many typos. For instance, "a another" (page 2); "motion, [10,11]." (page 2); "permits and permits dissociation" (page 3); the citation to references 8 and 21, which are two of the most important references on the light-induced conical intersection, are not correctly cited as "J. Chem. Phys. 139, (2013)" and "Phys. Rev. Lett. 116, 1 (2016)" (pages 20 and 21); and so on that I cannot completely list here.

We would like to thank the reviewers for taking the time to thoroughly review our manuscript. We believe that their constructive criticism has helped us to improve the manuscript. Below we respond to each referee's comments in detail.

Reply to Referee #1:

Comment 1: *"In Fig. 3 (page 12) the proton momentum distributions obtained by solving the TDSE without and with rotation are shown. Here the black arrows indicate the population transfer through rotational dynamics. I would suggest to give some numbers to announce the magnitude of the total dissociation products. It would be very informative. It is true that there are modulations in the angular distribution even without including the rotation, but in spite of this, it seems that the dissociation products can be significantly different in the non-rotational and the 2D models."*

We agree with the referee that quantitative indication of a change between the 1D and 2D model is interesting. To some extent the color code achieves this goal. For example, the peak at $p_{VIS}=0$, $p_{IR}=6.5\text{au}$ becomes visibly stronger when rotations are permitted. To help further quantify the redistribution of the nuclear wavepacket due to the rotational motion one could normalize the distribution by its integral. However, even then the comparison of the actual redistribution path with and without rotations will remain somewhat speculative.

In order to hopefully satisfy the referee's request for a more quantitative evaluation of the difference in the angular emission patterns we have added an inset to figure 3 where we plot the radially integrated angular distributions from the calculations with and without rotations. We also added the experimental distribution to the inset. We believe that this plot helps to better assess the considerable impact of rotational dynamics to the final momentum distribution.

In the text on page 10 we have added:

"A direct comparison of the calculated and measured angular distributions is given by the inset of figure 3. Indeed, the strong modulations observed in the experimental data are only obtained in the simulations which take rotations into account. However, the modulation depth in the experimental data is smaller than in the simulations with rotations, which is ascribed to the reduced dimensionality of the simulations and the experimental resolution."

The following text has been added in the caption of Figure 3:

"The inset compares the calculated proton angular distributions to the measured one presented in Figure 2(b). Each angular distribution is normalized to its integral. The errorbars for the experimental data are of the same size as the symbols."

Comment 2: *"The expressions (on page 17) are confusing: the definition of $F(t)$ in Eq. 1 is missing; following there are two different formulas with the same $E(t)$ on left side."*

We have corrected Formula 2 and 3 such that $F(t)$ designates the total laser field, and an index is added to avoid multiple uses of $E(t)$.

Comment 3: *“The authors may wish to cite the following references [G. J. Halász et al, J. Phys. Chem. A 117, 8528 (2013); G. J. Halász et al, Phys. Rev. A 88, 043413 (2013)] using a similar theoretical framework and where the initial vibrational wavepacket is defined as the Franck–Condon distribution of the vibrational states of the ion.”*

We thank the referee for pointing the missing references out and apologize for the oversight. We have added the references on page 2.

Comment 4: *“Small typos should be corrected.”*

We have corrected any typos throughout the manuscript that we could find.

Reply to Referee #2:

Comment 1: *“I do not have major objections for the paper to be accepted for publication in Nature Comm.; however, the authors should present more convincing arguments: What is the real message presented in the paper? Is it that from the observation of fragment angular distributions in coincidence at different time delays they show that the photodissociation process is more complex than expected for such a simple system? I do not see that this allows us to “visualize” the LIP. On the contrary, this indicates that there are many non-adiabatic processes in between LIPs. In this sense, the LIP concept is not so useful. Comments should be added in the manuscript.”*

We thank the referee for giving us the opportunity to strengthen the message of our paper. Naturally, the main experimental observation is the unexpectedly strong angular modulation of the emitted protons. Based on the analysis of the experimental results and the theoretical studies, we draw our main conclusion, namely that the angular ion yield is strongly influenced by the angle dependence of the light-induced potentials, and shaped by the rotational dynamics around the light-induced single and multi-photon intersections.

Insofar, we believe that the light-induced potential picture is useful as it predicts the points where angle-dependent channel switching takes place, and where prominent light-induced point intersections are present. While our data indicates that there is a relationship between the proton angular distribution and the LIPs, we agree with the referee that there is no direct way to visualize the LIPs from the proton angular distribution.

We would like to emphasize that we did not intend to claim that the LIPs could be visualized by recording the proton momentum distributions. Recognizing that the referee might have understood such an intention from the wording used in the abstract, we have replaced

“Here, we demonstrate in theory and experiment that the full complexity of such light-induced potential energy surfaces can be uncovered. In H_2^+ , ...”

with

“Here, we demonstrate that multi-photon couplings can give rise to complex light-induced potential energy surfaces that govern molecular behavior. In the laser-induced dissociation of H_2 ”

Moreover, in the introduction, we have modified the text to read

“Here, we show in theory and experiment that LIPs featuring non-linear light-induced point intersections can result in strong modulations of the angular ion yield. Using a two-color pump-probe scheme allows us to probe and control the nuclear dynamics, as the underlying LIPs evolve in the laser field.”

In the Summary and Outlook (page 15), we have added:

“The LIP picture predicts where angle-dependent channel switching takes place, and where prominent light-induced point intersections are present.”

Comment 2: *“In addition, more information should be included considering that they work with angular momentum $J=0$ (as I understood) which implies no initial rotation of the molecule, but there is an isotropic distribution with respect to the laser polarization.”*

Indeed we make the assumption that the simulated H_2^+ molecule is in the rotational ground state. Since this is a single eigenstate we agree with the referee that there is no initial rotation. In the experiment, we probe an unaligned ensemble of H_2 molecules, that is, all molecular orientations are equally likely to be encountered by our laser pulses.

This assumption is motivated by the fact that the measured H_2^+ translational temperature is a few Kelvin. From Raman measurements of H_2 molecules in a supersonic expansion (Montero et al., Journal of Chemical Physics 125, 124301 (2006)) we estimate the rotational temperature to be $<30K$.

Since all temperatures in the jet scale nearly linear with stagnation temperature, $T_0=60 K$ in our case, we can adapt Fig. 3 in Montero et al. (Journal of Chemical Physics 125, 124301 (2006)) to our case by dividing with a factor 5. Also, as we are measuring more than a meter away from the nozzle the asymptotic value of rotational cooling has been attained. With a stagnation pressure of 1.4bar our rotational temperature is somewhat below the asymptotic value provided for a 1.28bar stagnation pressure of $175K/5=35K$. At 35K more than 90% of the H_2 molecules will be in the rotational ground state. We note that, since we use natural H_2 we have a 75%/25% mixture of ortho- and para- H_2 , which means that the rotational ground state of 75% of our molecules is actually $J=1$. However, the difference between $J=0$ and $J=1$ has very little impact on the final momentum distributions.

We have modified the text in the Methods on page 17 to read:

“The rotational degree of freedom was initialized to a thermal rotational distribution, with temperature chosen to be low enough such that only the rotational ground state is populated. The initial distribution of the molecular axis with respect to the laser polarization is isotropic, closely reflecting the experimental conditions.”

On page 16, we modified the text to read

“After recombination, the pulses are focused in the center of a Cold Target Recoil Ion Momentum Spectrometer (COLTRIMS) [57], where they intersect an ultrasonic jet of pre-cooled ($T=60K$) of neutral H_2 ”.

Comment 3: *“Technically, I do not understand how the LIPs are calculated from Floquet according to the equations in the section of Floquet states (by the way, equations should be numbered!). The authors say that $\Delta t = \tau/N$ but there is no value given for N . It seems that this equation goes beyond the rotating wave approximation (RWA), but then only allows us to obtain 2 LIPs. How the authors can get all LIPs that cross at different hw_{IR} values? (for instance, those that are shown in Fig. 1c with $\sigma_u - 3hw_{IR}$, etc.). There is something missing here.”*

We thank the referee for the comment. We added any missing equation numbers.

Also, on page 18, we have added $N=1024$ for the number of discretization points used to build the one-period propagator.

The referee is correct that our Floquet formalism goes beyond the rotating wave approximation. The RWA is closer to the formalism used in the LICl picture put forward by Cederbaum et al., which only allows for the computation of the first-order Floquet resonance, i.e., only the single photon coupling between two states.

The full Floquet (non-RWA) theory has been widely used in the literature since the first description of bondsoftening in 1990 (again see, e.g., Refs 18 and 34), and is indeed capable of capturing all the multi-photon resonances. For many people in the strong field community, the existence of this Floquet ladder is well-known and expected. We concur that for a general audience as the one of Nature Communications the construction of the ladder could be better described, and we are attempting this in the following. However, for a full derivation of the formalism we need to refer the reader to the literature, e.g., Chu and Telnov, Physics Reports 390 (2004) 1–131; P. Hänggi, Driven quantum systems, in: Quantum Transport and Dissipation, Wiley-VCH, Weinheim, 1998, Ch. 5, pp. 249–286; J. Bayfield, Quantum Evolution: An Introduction to Time-Dependent Quantum Mechanics. Wiley-VCH, August 1999, ISBN 0-471-18174-9.

Specifically, the referee asks how the Floquet formalism allows us to obtain more than two light-induced potentials. Briefly, the Floquet method allows us to obtain two distinct LIPs with opposite parity. These two Floquet surfaces $\varepsilon_\alpha(R)$ (eq. 7, Methods) already have all multiphoton couplings built in. The plotted surfaces are a periodic extension of the two distinct light-dressed states, i.e., the σ_g (σ_u) manifold is obtained by adding even numbers of photon energies (0, ± 2 , ± 4 , ...) to the calculated light-dressed σ_g (σ_u) states.

To be a bit more explicit, the existence of more than two states (generally referred to as the ‘ladder’ of Floquet states) arises as follows. The Floquet states $|F_\alpha(R, t)\rangle$ in eq. 7 are periodic with the period of the laser oscillations, and they carry a sub-cycle time dependence whenever multiphoton couplings are active/important/non-negligible. Consequently, the associated potential energy surfaces will also, in general, exhibit a sub-cycle time-dependence. However, the sub-cycle time-dependence can be expanded as a Fourier series to yield a set of time-independent potentials that characterize the system

$$e^{-i\varepsilon_\alpha(R)t}|F_\alpha(R, t)\rangle = e^{-i\varepsilon_\alpha(R)t} \sum_{n=-\infty}^{\infty} |f_\alpha^n(R)\rangle e^{-in\omega t} = \sum_n |f_\alpha^n(R)\rangle e^{-i(\varepsilon_\alpha(R)+n\omega)t}.$$

The ladder of Floquet states is then the $(\varepsilon_\alpha(R) + n\omega)$ energies of the Fourier expansion; there are indeed only two α , but these two states get shifted by $n\omega$, where n runs over all integers, thereby constructing a ladder of states.

We replaced the text in the methods section after equation 7 with:

“The Floquet states $|F_\alpha(R, t)\rangle$ are the eigenstates of $U(t, t + T; R)$,

$$U(t, t + T; R)|F_\alpha(R, t)\rangle = e^{-i\varepsilon_\alpha(R)t}|F_\alpha(R, t)\rangle, \quad (7)$$

where the $\varepsilon_\alpha(R)$ are the quasi-energies of the Floquet states $|F_\alpha(R, t)\rangle$. The Floquet states and quasi-energies are found directly by diagonalizing the 2×2 $U(t, t + T; R)$ matrix for each R . The Floquet states $|F_\alpha(R, t)\rangle$ are periodic with the period of the laser field, and exhibit a sub-cycle time dependence whenever multiphoton couplings are active. Consequently, the associated potential energy surfaces will also, in general, exhibit a sub-cycle time-dependence. The sub-cycle time-dependence can be expanded as a Fourier series to yield a set of time-independent potentials that characterize the system

$$e^{-i\varepsilon_\alpha(R)t} |F_\alpha(R, t)\rangle = e^{-i\varepsilon_\alpha(R)t} \sum_{n=-\infty}^{\infty} |f_\alpha^n(R)\rangle e^{-in\omega t} = \sum_{n=-\infty}^{\infty} |f_\alpha^n(R)\rangle e^{-i(\varepsilon_\alpha(R)+n\omega)t}. \quad (8)$$

The ladder of Floquet states is formed by the energies of the Fourier expansion, where the $\varepsilon_\alpha(R)$ get repeated and shifted by $n\omega$, forming an infinite ladder of time-independent potentials. These quasi-energies ($\varepsilon_\alpha(R) + n\omega$) are what is referred to as light-induced potentials in the main text.

Reply to Referee #3

Comment 1: *“The authors investigate the light induced conical intersection in the dissociative ionization of H₂ molecules by observing a modulated angular distribution of the protons. A pump-probe scheme is used, where the pump ionization created H₂⁺ is dissociated by the probe pulse. The modulations in the angular distributions of the strong-field photo-dissociation of H₂⁺ has been measured several years ago, for instance, prl 116, 143004 (2016), where a 30 fs pulse at 795 nm was used to dissociate a H₂⁺ beam. The produced H and H⁺ fragments were measured in coincidence using a time- and position-sensitive detector, similar to the present work. The authors claimed that they investigated the non-linear light-induced point intersections in dissociative ionization of H₂, resulting in strong modulations of the angular ion yield. The observation of the “non-linear light-induced point intersections” is indeed an addition to previous works, e.g. prl 116, 143004 (2016), but it is not fundamentally “new” and don’t stand for a significant advancement of this field by simply adding more photons. Based on the work of “single-photon” induced conical intersection, the multi-photon case is expected straightforwardly in a similar picture. For the very limited innovation, I cannot recommend the publication of the present paper in the highly impact journal of NC.*”

The referee has raised concerns on the degree of advance over previously published work. Namely, A. Natan, et al, PRL **116**, 143004 (2016) (Ref. 23 in our work) reported on the observation of modulations of the proton angular distribution as a compelling signature of a LIC. Importantly, the work in the present paper provides evidence that the situation is generally more complex, and that angular structures can both be induced by dynamics around a LIC and by the existence of multi-photon couplings. The importance of multiphoton couplings is evidenced by the fact that modulations similar to the experimental ones are already present in the simulations that neglect rotational dynamics. The agreement between theory and experiment is further improved by including rotations, as in the LIC case (Refs. 21 and 23).

Specifically, the topology and character of the linear and nonlinear light-induced intersections are different. In the world of non-adiabatic molecular dynamics, there is a clear distinction between a conical intersection (where potential energy surfaces diverge linearly as one moves away from the point of intersection) and higher-order crossings (such as Renner-Teller intersection, to give one example, where the potential energy curves diverge quadratically, or even higher order, as one moves away from the point of intersection). Each of these different intersections, where the degeneracy is lifted to different orders as one moves away from the point of crossing, have different spectroscopic signatures, consequences, and characteristics. Single-photon light-induced crossings (cf. Fig 1a) are analogous to the molecular conical intersections (in that they lift the degeneracy linearly as one moves away from the crossing), while the higher-order light-induced crossings (cf. fig. 1d) are not (here the degeneracy is lifted at a higher-order than linear). We have added a new figure and discussion in Supplementary Note 1 to elaborate on this point.

In the same way that field-free non-conical (higher-order) intersections in molecular physics are considered as structures distinct from conical intersection, higher-order light-induced crossing will also display properties distinct from the single-photon crossing, and they are deserving of their own focus and attention that goes beyond being a footnote to the single-photon crossing.

Since not only the experimental observations but also the physical picture to explain them significantly differs from previous work, we are convinced that our paper deserves publication in a high-impact journal such as Nature Communications.

Finally, in order to aid the comparison of the work by Natan, et al. to our present work, we summarize differences between them in the table below.

A. Natan, et al.	Our work
Single-photon couplings (LICI)	Single and multiphoton couplings (beyond LICI)
Angular modulations in narrow energy bins ($\Delta E = 25$ meV)	Strong angular structure that survives integration over kinetic energy
Modulations due to rotational scattering around the LICI	Modulations due to an interplay of angle-dependent channel switching and rotational dynamics
Data were symmetrized	No symmetrization was done
Starts with an incoherent sum of vibrational states in H_2^+	Starts with an coherent sum of vibrational states produced by strong-field ionization of neutral H_2
Single color 800nm	Two color 730nm + 2300nm
Static	Time-resolved (delay dependence)

Comment 2: *“In PRL 116, 143004 (2016), it was noticed that the modulated angular distribution depends on the vibrational levels of the initially populated H_2^+ , which will be smeared out after an integration over various vibrational levels. Is this the case in the present experiment? Do the authors resolve the vibrational levels of the pump ionization created H_2^+ ? I guess this effect may be more serious for the multiphoton induced conical intersection than the sing-photon process.”*

In our experiment, contributions from individual vibrational states are not resolved. In strong-field photo-dissociation/ionization experiments where the neutral H_2 molecule is ionized in the laser pulse, the vibrational levels are usually not resolved. The reason is that tunnel ionization of the neutral molecule creates the vibrational wavepacket in the molecular ion. In the presence of these high field strengths (high enough to overcome a 15eV ionization barrier) the nuclear wavepacket efficiently disperses, smearing the dissociation energies. On the other hand, in an experiment with H_2^+ precursor molecules from an ion source, subjected to relatively long, 30 fs pulses the adiabatic turn-on of the laser field guarantees a well-defined depletion of each vibrational state within the pulse envelope (see Fig. 3, Prabhudesai et al., PRA 81, 023401 (2010), and ref. 12-16).

However, contrary to PRL 116, 143004 (2016), in the present experiment, the angular modulations survive the integration over various vibrational levels, as can be seen directly in our Fig. 2. The vibrational level is determined by the KER, that is, the squared radial momentum of the protons. In Fig. 2 of Prabhudesai et al., PRA 81, 023401 (2010) a KER-resolved angular distribution is shown that probably resembles the raw data shown in ref. 23. We have converted our data into the same coordinates, and show them in Figure 1 below for comparison with the data from Prabhudesai et al.. Clearly, our modulations are largely independent within a KER range that covers several vibrational levels. Note that the lower kinetic energies in our experiment result from the lower photon energy of the 2300 nm field, compared to the 800 nm light used by Prabhudesai et al.

Figure 1 Comparison between results from Prabhudesai, et al., using an ion beam source (left), and the present experiment (right).

In the text, we have added on page 7:

“Moreover, the angular structure survives averaging over kinetic energy, in contrast to the weaker modulations in previous work on LICl [23].”

Comment 3: “I am a little confused by the “role” of the rotational dynamics. The authors claim that the point section or LICl promotes the rotational dynamics. However, they also claim that the rotational dynamics plays “secondary role in defining the final momentum distribution”. The agreement between of the simulation results shown in figure 3(a) is better when the rotational dynamics is switched off. The simulated splitting around +/-10 degree is not observed in the experiments shown in figure 2. ”

In Figure 3 of the paper, we have added an inset that compares the momentum-integrated proton angular distribution for both simulations (with / without rotations) to the experimental data presented in Figure 2 of the paper. It can be seen there that a splitting at small angles is clearly observed in the experiment. We agree, however, that the splitting is not observed in the feature at larger momenta (~ 7 a.u.). Nevertheless, the simulations without rotations predict a much narrower feature than observed experimentally. Moreover, the magnitude of the measured angular modulations is significantly larger than the simulated ones if rotations are neglected. On this basis, we would tend to disagree with the reviewer’s assessment that the simulations without rotations exhibit better agreement with the experimental data.

At the end of page 10 / beginning of page 11, we added the following statement to summarize the mechanism creating the angular modulations in the absence of rotations:

“In the absence of rotations, the angular structure arises through a process we shall call angle-dependent channel-switching, as different orders of multiphoton couplings dominate at different alignment angles of the molecular axis with respect to the laser polarization.”

Comment 4: “For the results presented in figure 3, is the delay set to be zero between the few-cycle visible pulse and the IR pulse?”

A simulation of the delay dependent angular distribution of the protons, similar to figure 4, will be helpful for one to understand the experimental observations.”

The simulation results are integrated over a delay range corresponding to one optical cycle of the mid-IR field around the center of its envelope and integrated over the CEP of the visible pulse, as described in the Methods. In the caption of figure 3, we added:

“The calculations were performed for delay values corresponding to one IR cycle around the temporal overlap between the few-cycle visible pulse with the center of the mid-IR pulse, and integrated over one mid-IR cycle.”

We have added a figure with the suggested additional calculations in the SI, and added the following sentence on page 14 of the manuscript, referring to the new Supplementary Figure:

“Calculated proton momentum distributions for different delay values, which are consistent with these conclusions, are presented in Supplementary Figure 6.”

Comment 5: *“As shown in figure 4, the angular distribution of the protons strongly depends on the time delay between the pump and probe pulses. For dissociation of the H₂⁺ created by the pump pulse, the LICI induced by the probe is expected to be more significant when the pump pulse coming earlier, and should also be clearly observed even there is no temporal overlap between the pump and probe, i.e. the pump is much advanced than the probe. This will be similar to previous observations in prl 116, 143004 (2016) where a H₂⁺ beam is used. The authors need to confirm this point in their experiments. This also relates to the physical mechanism of their observations, and will be helpful to exclude other effects which require the temporal overlap of the pump and probe pulses.*”

The referee raises the interesting question, whether a transition from the present two-color LIP scenario to the conventional LICI case can be achieved by separating pump and probe pulses in time.

Notably, the splitting of the signal around 0° is already observed when the visible pulse arrives early during the mid-IR pulse (i.e., 40fs before the intensity maximum), as shown in Fig. 4(b) of the paper. Figure 2 below shows that for even earlier delays, 60 fs before the intensity maximum, the feature at intermediate angles disappears, but the ones around 10° remain.

Figure 2 Proton momentum distribution recorded for time delays where the visible pulse precedes the center of the IR pulse by 60±5 fs.

Unfortunately, experimental data where the visible pulse is much advanced with respect to the mid-IR pulse is not available for cross-polarized visible and mid-IR pulses.

Figure 3 below shows data for parallel polarization of the two pulses, where the visible pulse precedes the center of mid-IR pulse by more than 200fs. As for the data presented in Fig. 2(c) of the paper, no significant angular modulations are observed.

Figure 3 Proton momentum distribution recorded with parallel polarization of visible and IR pulses for the case where the visible pulse precedes the center of the IR pulse by more than 200 fs.

Hence, on the basis of existing data we could not directly observe the transition between the two regimes mentioned above. Instead, the additional results presented here instead confirm the conjecture (made on page 7 of the manuscript) that the pronounced angular structure arises from the two-color laser field used in our experiment.

Figure 4 TDSE simulations for separated pump and probe pulses. (a) The electric field for delay value of 150fs showing the horizontally polarized visible few-cycle pulse with pedestal, and the vertically polarized mid-IR pulse. The calculated proton momentum distributions are presented for two cases: (b) in the absence of the IR pulse and (c) for an IR peak intensity of $3 \times 10^{13} \text{ Wcm}^{-2}$. The normalized difference of the two momentum distributions (d) shows only weak differences, except for the signal along the IR polarization. The enlarged view (e) reveals weak variations in the angular distributions along the visible polarization.

In Figure 4, we present numerical results for separated visible and IR pulses. As shown in Figure 4, (a), the visible pulse pedestal has vanished by the time the IR pulse arrives. By calculating the momentum distribution in the absence of the IR pulse and in its presence, we can directly probe its effect on the angular distribution of the protons. Weak variations of few percent are observed in the dissociation signal originating from the visible pulse. Closer analysis of these variations is beyond the scope of the present paper. We will address them in an upcoming study, for which two-color experiments using an ion beam source will be conducted. This will allow for closely investigating the relationship between the two regimes.

Comment 6: *“A pump pulse of 5 fs was used in the experiment to ionize H2. For such a short laser pulse, I am wondering the probability of the dissociation of the self-ionization created H2+ within the same pulse. Let’s assume that the ionization probability is maximal around the peak of the 5 fs pulse, and the molecular ion will not have enough time to stretch to the critical internuclear distance for the one-photon dipole transition in the falling edge of the 5 fs pulse. More explanation is required here for the observation of pronounced signal along the polarization of the pump pulse. Is it because of the existence of the pedestal of the few-cycle pulse? If this is the case, an experimental measurement of this pedestal is necessary for its important role for the experimental results.”*

We agree with the referee that a “clean” 5fs pulse (i.e. one without a pedestal or post pulses) is expected to cause only little dissociation by bond softening. However, few-cycle pulses obtained from hollow-core fiber compression are usually not clean, see e.g. KT Kim, et al, Nat. Photon. 7, 958 (2013) (Supplementary Reference [1]). This leads to notable bond softening by the visible 5-fs pulse alone, as shown in the bottom inset of Figure 2b. Note that this result agrees well with earlier studies using few-cycle pulses, such as M. F. Kling, et al., Science 312, 246 (2006), where a

pronounced bond softening signal was observed even in D_2 , where nuclear dynamics are considerably slower than in H_2 .

While we agree that a quantitative measurement of the pulse pedestal present in our experiment would be a reasonable addition to the presented data, the existence of the pedestal is out of question. In the absence of a quantitative measurement of the pulse pedestal, we present below a non-linear autocorrelation signal obtained in a separate pump-probe experiment, recording the strong-field ionization yield from N_2O . Because of the undefined non-linearity of this process, a retrieval of the pulse and pedestal duration is not straight-forward.

Figure 5 Non-linear autocorrelation signal for a pair of identical, visible few-cycle pulses. The data was obtained in the strong-field ionization from N_2O .

The data presented in Figure 5 shows only few strong oscillations, indicative of the few-fs pulse duration of our visible pulses. Clear oscillations are observable up to approximately $\Delta t = 50$ fs, which is consistent with the pedestal duration used for the TDSE simulations. Furthermore, this duration of the pulse pedestal and its intensity are in agreement with previously published experimental data (KT Kim, et al, Nat. Photon. 7, 958 (2013)).

To further justify the assumptions made for the pedestal in our calculations, we have modified the text on page 9 to read

“For the visible few-cycle pulse, we also consider a weak pulse pedestal at 5% of the peak intensity and with a duration of 45 fs (full width at half-maximum of the intensity envelope). These values are consistent with field-resolved measurements of few-cycle pulses [53]. The initial alignment of the molecular axis with respect to the laser polarization is assumed to be isotropic.”

Comment 6: *“The writing of the paper is not carefully checked with many typos. For instance, “a another” (page 2); “motion, [10,11].” (page 2); “permits and permits dissociation” (page 3); the citation to references 8 and 21, which are two of the most important references on the light-induced conical intersection, are not correctly cited as “J. Chem. Phys. 139, (2013)” and “Phys. Rev. Lett. 116, 1 (2016)” (pages 20 and 21); and so on that I cannot completely list here”*

Thank you for pointing out these mistakes. We have corrected these and any other typos we found.

Further changes

1. We have added a new Reference [30], J. F. McCann and A. D. Bandrauk, Phys. Rev. A 42, 2806 (1990) and refer to it in the introduction on top of page 3 by adding

“and anomalous fragment angular distributions have been predicted in the non-perturbative regime [30].”

2. The last sentence of the introduction has been expanded to read

“While these features are absent in the single-photon coupling regime at low intensity, significantly higher intensities produce very convoluted dissociation patterns that involve high-order couplings but would likely defy experimental resolution, see Supplementary Figure 2.”

Reviewers' comments:

Reviewer #1 (Remarks to the Author):

The authors have answered all my questions and they have improved the manuscript following the recommendations. I think that this manuscript can now be published in Nature Communications.

Reviewer #2 (Remarks to the Author):

In my view the authors have addressed all the suggestions/objections by the reviewers in a satisfactory manner. Therefore, I recommend the manuscript to be accepted for publication in Nature Comm.

Reviewer #3 (Remarks to the Author):

I would like to thank the authors for the detailed reply to my comments and the revision of the manuscript. I would like to say that the experimental data is very beautiful and really impressive, but the physical explanation is improper, i.e. not the high order crossing of the potential energy surfaces induced by the IR pulse. I would like the authors to carefully consider my following comments.

According to the physical picture of the nonlinear light-induced point intersection, in analogy to the single photon induced conical intersection, the ionization by the VIS pulse creates a molecular ion, which is afterwards dissociated on the perturbed potential energy surfaces induced by a time delayed IR pulse via a multiphoton coupling, leading to modulated angular distribution of the ejected protons. This is the motivation of the authors in designing the experiments, I guess. In this physical picture, no temporal overlap between the pump VIS and probe IR pulses is required. Any temporal overlap will change the laser waveform, although the IR is much weaker than the VIS, and thus its interaction with the molecular ion and modulate the dissociation process.

The main result of the present manuscript is the very beautiful modulation of the angular distribution of the ejected protons as shown in figure 2b, where the VIS pulse sits at the center of the IR pulse. I guess the strongly modulated structure of the inner ring will vanish if there is no temporal overlap between the IR and VIS pulses, e.g. the IR pulse is time delayed with respect to the VIS pulse. It is the scheme that the higher order crossings are expected to be probed as the authors claimed in their manuscript "The intent of the two-pulse scheme is to decouple the production of the molecular wavepacket from the field that generates the LIPs."

The use of a weak visible field corresponding to the pedestal of the few-cycle VIS pulse is also very strange in the two-color Floquet theory. I guess the results will complete different if the pedestal is removed, which should not be there for the physical picture of the high order crossing induced by the IR probe pulse. The modulation of the crossing of the potential energy surfaces and thus the channel switching described on page 11 of the manuscript is resulted from the modulation of the laser waveform due to the interference of the temporally overlapped IR and VIS pulses.

It is very clear that the using of the "recoil frame" by removing the impact of the electron recoil helps to increase the visibility of the modulated structure in figure 2b, by comparing the results in supplementary figure 6. If I understand correctly, the momentum of the electron from the pump VIS pulse (which does the ionization) should have a cigar shaped distribution along the horizontal direction. The recoil of this electron will give a backward recoil momentum to the ion along the horizontal direction. However, if we carefully compare the laboratory frame and recoil frame data in supplementary figures 6a and 6b, the recoil correction seems mainly affect the distribution along

the vertical axis or an angle from it. It indicates a complex momentum distribution of the freed electron by the pump pulse, or it is resulted from the combined effect of the VIS and IR pulse when they temporally overlap. How does the 2D momentum distribution of the correlated electrons look like? I am wondering the results presented in figure 2b (the main result of the present work) is mainly due to the modulated waveform of the laser field when the VIS and IR pulses are temporally overlapped, although the IR is much weaker than the VIS, rather than the "high order crossings" of the potential energy surfaces of the molecular ion. This "high order crossings" of the molecular ion potential surfaces should be probed in a pump-probe scheme where the VIS is temporally advanced with respect to the IR without temporal overlap. It also explains why the predicted results in figure 1e (solely induced by the IR pulse for the really physics of the high order crossing) significantly differs from the experimental data in figure 2b.

Therefore, I suggest to present the experimental or simulation results for a clean VIS pulse without pedestal, which will remove the effect induced by the modulated laser waveform and give one a clean picture of the high order crossings induced by the IR pulse.

In brief, my feeling is that the observations presented here more or less is due to the modulation of the laser waveform of the temporally overlapped IR and VIS pulses, rather than the high order effect of the crossing potential energy surfaces induced by the IR pulse. It can be clarified by cancelling the temporal overlap between the pump VIS pulse and the probe IR pulse.

We would like to thank all referees for reviewing our revised manuscript. Referees #1 and #2 have accepted our revisions and recommend publication. In the following we will address the remaining issues raised by Referee #3.

"I would like to say that the experimental data is very beautiful and really impressive, but the physical explanation is improper, i.e. not the high order crossing of the potential energy surfaces induced by the IR pulse. I would like the authors to carefully consider my following comments."

We thank the referee for the nice words about our experimental results and we recognize that the referee has not been convinced by our explanation of these results. However, we believe that, after careful consideration of the referee's comments, we can reconcile our explanation and the referee's picture of the physical mechanism underlying the experimental observations.

Physical interpretation

The referee's main concern is that "the observations [...] is due to the modulation of the laser waveform of the temporally overlapped IR and VIS pulses, rather than the high order effect of the crossing potential energy surfaces induced by the IR pulse".

In detail, the referee writes:

"According to the physical picture of the nonlinear light-induced point intersection, in analogy to the single photon induced conical intersection, the ionization by the VIS pulse creates a molecular ion, which is afterwards dissociated on the perturbed potential energy surfaces induced by a time delayed IR pulse via a multiphoton coupling, leading to modulated angular distribution of the ejected protons. This is the motivation of the authors in designing the experiments, I guess. In this physical picture, no temporal overlap between the pump VIS and probe IR pulses is required. Any temporal overlap will change the laser waveform, although the IR is much weaker than the VIS, and thus its interaction with the molecular ion and modulate the dissociation process.

The main result of the present manuscript is the very beautiful modulation of the angular distribution of the ejected protons as shown in figure 2b, where the VIS pulse sits at the center of the IR pulse. I guess the strongly modulated structure of the inner ring will vanish if there is no temporal overlap between the IR and VIS pulses, e.g. the IR pulse is time delayed with respect to the VIS pulse. It is the scheme that the higher order crossings are expected to be probed as the authors claimed in their manuscript 'The intent of the two-pulse scheme is to decouple the production of the molecular wavepacket from the field that generates the LIPs.'"

After two more specific questions, which are addressed below, the referee returns to the discussion of the physical interpretation:

"I am wondering the results presented in figure 2b (the main result of the present work) is mainly due to the modulated waveform of the laser field when the VIS and IR pulses are temporally overlapped, although the IR is much weaker than the VIS, rather than the "high order crossings" of the potential energy surfaces of the molecular ion. This "high order crossings" of the molecular ion potential surfaces

should be probed in a pump-probe scheme where the VIS is temporally advanced with respect to the IR without temporal overlap. It also explains why the predicted results in figure 1e (solely induced by the IR pulse for the really physics of the high order crossing) significantly differs from the experimental data in figure 2b.

Therefore, I suggest to present the experimental or simulation results for a clean VIS pulse without pedestal, which will remove the effect induced by the modulated laser waveform and give one a clean picture of the high order crossings induced by the IR pulse.

In brief, my feeling is that the observations presented here more or less is due to the modulation of the laser waveform of the temporally overlapped IR and VIS pulses, rather than the high order effect of the crossing potential energy surfaces induced by the IR pulse. It can be clarified by cancelling the temporal overlap between the pump VIS pulse and the probe IR pulse.”

Figure I Calculated proton momentum distribution arising from the photodissociation of H_2^+ by a 45-fs mid-IR (2300 nm) pulse at $3 \cdot 10^{13} \text{ Wcm}^{-2}$ (a) only, (b) in combination with a 5-fs visible pulse, (c) in combination with a 7-fs visible pulse. The visible pulse has a carrier wavelength of 730 nm and a peak intensity of $2 \cdot 10^{14} \text{ Wcm}^{-2}$.

In Figure I, we present the results of our TDSE simulations for various pulse durations of the visible pulse with the pedestal of the visible pulse removed, as requested by the referee. In Figure I (a) a “zero fs” VIS pulse duration reproduces the results show in Figure 1 (e) of the manuscript. Figure I (b, c) shows the proton momentum distributions obtained with the IR pulse and a clean few-cycle visible pulse of two different pulse durations with both pulse envelope maxima coinciding. Evidently, the dissociation dynamics are affected by a 5-fs pulse, while a 7-fs pulse already produces proton distributions that are reminiscent of our experimental results. This shows that probing the LIPs resulting from the IR field only, in fact, requires extremely short and clean pulses, as discussed in the Summary and Outlook section on page 16:

“...Probing the LIPs produced by the mid-IR dressing field on its own may be improved by using a shorter pulse for preparation of the bound wave packet, such as a few-cycle UV or attosecond pulse. Previous experiments along these lines (e.g., [44], [57]) were conducted in the single-photon dressing regime and did not study the influence of the light induced potential surfaces on the angular dependence of dissociation.”

Thus, we completely agree with the referee that our experimental observations can only be explained by the simultaneous action of both laser fields. However, we think that this is not necessarily in contradiction with our claim that our experiment probes high-order Floquet states. On the contrary, the

reasons for the access to high-order point-intersections, i.e., the relatively low intensity and long wavelength, continue to remain operative even in the presence of a second, perpendicularly polarized light field as the structural progression in the proton spectra in Figure I clearly shows. Thus, the nuclear wavepacket propagates on the combined two-color high-order potential energy surfaces in Fig. 3c instead of those in Fig.1d.

In our manuscript, the fact that both laser fields contribute to the dissociation process is demonstrated already from the experimental data alone. In Fig. 2, we compare the proton momentum distribution obtained with perpendicularly polarized two-color pulses (Fig. 2b) to the results obtained with two-color pulses with parallel polarization (Fig. 2c). From this comparison, the conclusion is drawn that the visible pulse must be considered in the further analysis. In order to express this conclusion more clearly, we have replaced on page 7

“Although the intention of our scheme was to decouple the effects of the mid-IR and visible pulses, the striking dependence of the angular structure on the relative polarization of the two laser fields implies that the visible field contributes to the formation of the additional spots and must be considered in further investigations.”

with

“Although the intention of our scheme was to decouple the effects of the mid-IR and visible pulses, the striking dependence of the angular structure on the relative polarization of the two laser fields implies that the visible field contributes to the formation of the additional spots. It will thus be considered in the following analysis of our results.”

We acknowledge that the importance of both laser colors in the dissociation process may have been mentioned somewhat late in the manuscript. Therefore, we have made adjustments to the abstract, where we replaced

“In the laser-induced dissociation of H_2^+ , the simplest of molecules, we observe a strongly modulated angular distribution of protons which has escaped prior observation. These modulations directly result from ultrafast dynamics on the light-induced molecular potentials and can be modified by varying the amplitude, duration and phase of the mid-infrared dressing field. This opens new opportunities for manipulating the dissociation of small molecules using strong laser fields.”

with

“In the laser-induced dissociation of H_2^+ , the simplest of molecules, we measure a strongly modulated angular distribution of protons which has escaped prior observation. Using two-color Floquet theory, we show that the modulations result from ultrafast dynamics on light-induced molecular potentials. These potentials are shaped by the amplitude, duration and phase of the dressing fields, allowing for manipulating the dissociation dynamics of small molecules.”

In addition, we now mention the role of both laser colors in the introduction, where we added on page 3:

“The experimental results are interpreted with the help of numerical solutions of the time-dependent Schrödinger equation and two-color Floquet theory. We find that the two-color laser field gives rise to remarkably complex dissociation dynamics that underlie our experimental results.”

Furthermore, we have modified the text in the summary section on page 16 to read:

“We have shown that the modulations can be understood as signatures of complex light-induced potential energy landscapes that are shaped by both single-photon and multiphoton transitions in a cross-polarized two-color laser field.”

Finally, we have added the above figure and discussion to the supplemental material as Supplementary Figure 8 in Supplementary Note 6.

Two-color floquet theory:

“The use of a weak visible field corresponding to the pedestal of the few-cycle VIS pulse is also very strange in the two-color Floquet theory. I guess the results will complete different if the pedestal is removed, which should not be there for the physical picture of the high order crossing induced by the IR probe pulse. The modulation of the crossing of the potential energy surfaces and thus the channel switching described on page 11 of the manuscript is resulted from the modulation of the laser waveform due to the interference of the temporally overlapped IR and VIS pulses.”

We agree with the referee on all points.

Just to be clear: the Floquet potentials shown in Fig 1 d and 3 c have been calculated with plane waves, and not with time-dependent pulses, nor have we propagated nuclear wavepackets on the Floquet potentials. They are really only for illustration purposes.

Considering the VIS field in the Floquet calculations is motivated by the experimental evidence that both pulses together significantly affect the dissociation process, and that effect is strongest when the VIS pulse overlaps with the envelope maximum of the IR pulse.

Of course, the Floquet results are very different if the VIS field is removed. This case is shown in Fig. 1d, which is quite different to the two-color Floquet states shown in Fig. 3c). The channel switching is a result of the modulations in the potential energy surfaces, which are more pronounced in the case of the two-color Floquet states.

The pedestal of our VIS pulse on the other hand is dictated by our experimental imperfection in generating the few cycle pulse. The TDSE calculations in Fig 1 and 3 naturally include all couplings, and there the agreement between theory and experiment is improved when the pedestal is included in the simulations.

Photoelectron momentum and recoil frame:

“It is very clear that the using of the “recoil frame” by removing the impact of the electron recoil helps to increase the visibility of the modulated structure in figure 2b, by comparing the results in supplementary figure 6. If I understand correctly, the momentum of the electron from the pump VIS pulse (which does the ionization) should have a cigar shaped distribution along the horizontal direction. The recoil of this electron will give a backward recoil momentum to the ion along the horizontal direction. However, if we carefully compare the laboratory frame and recoil frame data in supplementary figures 6a and 6b, the recoil correction seems mainly affect the distribution along the vertical axis or an angle from it. It indicates a complex momentum distribution of the freed electron by the pump pulse, or it is resulted from the combined effect of the VIS and IR pulse when they temporally overlap. How does the 2D momentum distribution of the correlated electrons look like?”

Figure II shows the 2D lab-frame momentum distribution of the photoelectrons detected in coincidence with the protons. Clearly, the mid-IR field has a strong impact on the final electron momentum distribution, even though it is much weaker than the VIS field. Due to the quadratic scaling of the ponderomotive potential with wavelength, the electron can pick up approximately as much energy from the IR field as from the much stronger VIS field. In addition to that, the photoelectron momentum along the VIS polarization is strongly peaked around zero, because ionization is closely confined to times around the peaks of the VIS pulse. On the other hand, ionization can occur at any phase of the IR pulse (depending on the delay between the two pulses), which leads to broad distribution along p_{IR} , (cf. Supplementary Fig. 5). The vertical axis in Supplemental Figure 5 denotes the electron momentum along the IR field polarization. The distribution shows the delay-dependence of the recoiling electron momentum. Since the data in Fig 2b) of the manuscript is integrated over several cycles of the IR the lab-frame proton momentum distribution is mostly blurred along p_{IR} .

We have changed the vertical axis label of Supplemental Figure 5 to be consistent with the rest of the manuscript.

Figure II. Measured photoelectron momentum distribution in the polarization plane. The projection on the two axes are shown on the right and on the bottom, respectively. The data has been integrated over the delay range [-15,15] fs, as Figure 2b of the main text. Note that the photoelectron detector has no resolution along p_{VIS} for $p_{\text{IR}} \sim 2$ a.u.

In conclusion, we hope to have been able to convince the referee that our understanding of the physical mechanism behind the experimental observations does not diverge from the referee's picture; the two-color field gives rise to the remarkable structure, and the nuclear wavepacket propagates and diffracts on this two-color induced two-dimensional potential energy landscape with conical and higher order, non-linear point intersections.

Reviewers' comments:

Reviewer #3 (Remarks to the Author):

Comments for authors:

It seems that the authors get the key point I concerned in my previous report. They agree with my comment that the data presented here is mainly resulted from the combined effect of the temporally overlapped VIS and IR pulses. Therefore, the results cannot be simply explained as the laser-induced high-order conical intersections of the potential energy curves, which should be observed by a single intense IR pulse. This is the initial motivation of the manuscript. To demonstrate the effect of this high order crossing, a pump-probe two-pulse scheme is proposed to decouple the production of the molecular wave packet by the ionization pulse from the dissociation pulse that generates the high-order conical intersections (see the description on page 6 in the section of "Structured proton angular distribution"). However, in facts, both in the simulation and experiment, the pump and probe pulses were temporally overlapped. Therefore, the results cannot support the physical picture of the high-order conical intersections.

I would suggest the authors to change the motivation or arguments of the paper, i.e. removing the high order crossing of the potential energy curves (conical intersections) with respect to the single photon induced conical intersection. It will require a lot of work, but will clarify the physical picture or a new story of the experimental observations. A complex laser waveform will be formed by using two temporally overlapped laser pulses of different polarizations and wavelengths. This complex waveform will induce rich dissociation dynamics depending on the details of the laser field or the relative phase and strength between the two pulses. This is clearly observed in the present paper.

Only if the authors can observe similar results for the case that the probe IR pulse is delayed with respect to the pump VIS pulse without any temporal overlap, they can keep the present explanation. In this physical picture, an ultrashort VIS pulse ionizes the H₂ into H₂⁺, and a time delayed (without any temporal overlap) IR pulse dissociates the H₂⁺ via the probe-induced high-order conical intersections of the potential energy curves.

As I mentioned in my previous report, the observations presented here is mainly due to the modulation of the laser waveform of the temporally overlapped IR and VIS pulses, rather than the high order effect of the crossing potential energy surfaces induced by the IR pulse which does not require any temporal overlap of the VIS pulse.

In the following we will address the report by Referee #3.

Referee 3: "It seems that the authors get the key point I concerned in my previous report. They agree with my comment that the data presented here is mainly resulted from the combined effect of the temporally overlapped VIS and IR pulses."

Yes, we agree that the modulations in the angular distributions are a result from the combined effect of the temporally overlapped VIS and IR pulses. However, note, that this statement does not per se exclude the existence of multiphoton couplings in these overlapped two color fields. In other words, there is neither a logical nor a compelling argument that the temporal overlap between the two fields would preclude the effect of high-order non-conical intersections on the dissociation dynamics.

Before we go on, we believe it is important to reiterate some fundamental concepts which seem to be often confused by the referee: high-order (point) intersections are *not conical* by definition. An intersection in general is a point in molecular geometry where potential energy surfaces (PES) are degenerate. The term conical means that the energy splitting between the two involved PES depends *linearly* on a coordinate, often a bending angle or a stretch motion.

In light-induced *conical* intersections of H_2^+ this coordinate is the angle between the molecular axis and the field: the field-mediated coupling strength between the $1s\sigma_g$ and the $2p\sigma_u$ increases approximately linearly around an angle of 90 degrees. Such a light-induced *conical* intersection dominates for rather weak fields where multiphoton processes are not relevant. In experiments performed on H_2^+ from ion sources such as the one by Natan et al.¹ (ref. [23] in the manuscript), the laser field turns on relatively slowly (compared to the vibrational period) and therefore in these experiments the light-induced 1-photon *conical* intersection is dominant in the dissociation spectrum of H_2^+ .

In high-order *non-conical* intersections or multiphoton couplings, this coupling strength increases *non-linearly* as a power law with the order of photons involved. This is directly visible (and indicated) in Fig. 1d) in our manuscript. As per our arguments above, the terms *high-order*, *non-linear* and *non-conical* are synonymous in the context of intersecting PES.

A *point* intersection is the intersection of two PES in a single point. Of course, there can be intersecting seams and more generally intersecting hypersurfaces, if the dimensionality of the system is high enough. For something as simple as H_2^+ the light-induced intersections of its PES are simply zero-dimensional points.

Referee 3: "Therefore, the results cannot be simply explained as the ling-induced high-order conical intersections of the potential energy curves, which should be observed by a single intense IR pulse. This is the initial motivation of the manuscript. To demonstrate the effect of this high order crossing, a pump-probe two-pulse scheme is proposed to decouple the production of the molecular wave packet by the ionization pulse from the dissociation pulse that generates the high-order conical intersections (see the description on page 6 in the section of "Structured proton angular distribution")."

¹ A. Natan, M. R. Ware, V. S. Prabhudesai, U. Lev, B. D. Bruner, O. Heber, and P. H. Bucksbaum, Phys. Rev. Lett. **116**, 143004 (2016).

As discussed in the previous response letter, we agree that the experimental observations cannot be explained by the non-linear intersections of the one-color light-induced potentials by the IR field, but by those of the two-color non-linear light-induced potentials by both the IR and VIS fields.

For the purpose to “experimentally probe the light-induced molecular potentials depicted in Fig. 1(d,e)...”, we perform the experiment using a two-pulse scheme. This is motivated by

(i) the required intensity being in the “intermediate regime” where multiphoton couplings are present but the dissociation pattern is not too convoluted. At these intensities, H_2 is not efficiently ionized by the mid-IR field. Hence, we use a short VIS pulse to produce the molecular ion

(ii) the possibility to place the molecular ion into the field at a chosen point in time during the IR envelope. (“This allows for probing the LIPs at selected times within the mid-IR pulse by scanning the time delay between the laser pulses”).

In order to emphasize the importance of (ii), we present additional computational results in Fig. R1 for two cases. In the first case (a) the simulation is initialized in the center of the mid-IR pulse. This is the situation for which the results shown in Fig. 1(e) of the manuscript were obtained. In the second case (b), the simulation is initialized at the onset of the mid-IR pulse. This situation is similar to the experiment proposed by the referee where the pump and probe pulses are separated in time. It is further similar (but not identical) to performing the experiment with in ion beam apparatus that produces H_2^+ ions, as has been done in the work by Natan, et al., Ref. [23].

Figure R1. Dissociation of H_2^+ when the molecular ion is created (a) in the center or (b) at the onset of a mid-IR pulse (35 fs, $3 \cdot 10^{13} \text{W/cm}^2$, 2300 nm). Shown are (a,b) the electric field experienced by the molecular ion, and the proton momentum distribution for molecules dissociating on the σ_g (a.1, b.1, respectively) or σ_u (a.2, b.2, respectively) surface, as well as (a.3, b.3, respectively) the total dissociation signal.

As can be seen in Fig. R1 (a) and (b), the results for the two cases are quite different. In particular, in Fig. R1 (b), a very convoluted pattern is produced that is strongly aligned along the laser polarization. In a

realistic experimental scenario, such a convoluted pattern would appear largely featureless such that non-linear features cannot be unambiguously identified. This comparison highlights the importance of the pump-probe scheme in order to study the effect of multiphoton light induced potentials. Fig. R1 is included in the new version of the supplementary information (Supplementary Figure 9).

In our experiment, we obtain strongly structured proton angular distributions. It turns out that these experimental results strongly differ from the simulated distributions plotted in Fig. 1(e). Our analysis reveals that it is necessary to take into account the effect of the VIS pump pulse. Hence, we extend our theoretical work, both TDSE and Floquet theory, to take into account the VIS field. We find that the LIPs in the two-color case differ significantly from the one-color case. Nevertheless, the non-linear character of certain features in our results remain. This is evidenced by

1. the theoretical results presented in Fig. 3 of the manuscript, and the additional lineouts presented in Supplementary Figure 4
2. the delay-dependent experimental results presented in Fig. 4 of the manuscript
3. the pulse-length progression presented in Fig. S8 in the supplementary information, and in the previous response letter.

Referee 3: "However, in facts, both in the simulation and experiment, the pump and probe pulses were temporally overlapped. Therefore, the results cannot support the physical picture of the high-order conical intersections."

In the previous point we have motivated why the pump and probe pulses need to be overlapped in order to obtain a clear signal from the non-linear transitions. We have discussed in the paper, in the previous responses, and above that our results can be explained by the multiphoton ("high-order") two-color LIPs.

In short, there is no dichotomy between multiphoton transitions and the presence of more than one laser color.

Referee 3: "I would suggest the authors to change the motivation or arguments of the paper, i.e. removing the high order crossing of the potential energy curves (conical intersections) with respect to the single photon induced conical intersection. It will require a lot of work, but will clarify the physical picture or a new story of the experimental observations. A complex laser waveform will be formed by using two temporally overlapped laser pulses of different polarizations and wavelengths. This complex waveform will induce rich dissociation dynamics depending on the details of the laser field or the relative phase and strength between the two pulses. This is clearly observed in the present paper."

Because there is ample evidence that our experiment probes multiphoton light-induced potentials, we see no reason to change motivation or arguments of the paper.

Nevertheless, we appreciate the referee's suggestion to comment on the complexity of a crossed-polarized two-color laser field. We have, thus, added the following sentence in the introduction on page 3:

“Notably, the dissociation dynamics are sensitive to the rather complex shape of orthogonally polarized two-color fields [51].”

Moreover, on page 3 we have replaced

“We find that the two-color laser field gives rise to remarkably complex dissociation dynamics that underlie our experimental results.”

with

“We find that, owing to both linear and non-linear transitions as well as rotational dynamics, the two-color laser field gives rise to remarkably complex dissociation dynamics that produce the strong modulations in the angular ion yield.”

Referee 3: “Only if the authors can observe similar results for the case that the probe IR pulse is delayed with respect to the pump VIS pulse without any temporal overlap, they can keep the present explanation. In this physical picture, an ultrashort VIS pulse ionizes the H₂ into H₂⁺, and a time delayed (without any temporal overlap) IR pulse dissociates the H₂⁺ via the probe-induced high-order conical intersections of the potential energy curves.”

As we have shown numerically in Fig. R1 (b), the proton momentum distributions for the case of separated pump and probe pulses do not allow for identifying the signatures of the non-linear LIPs shown in Fig. 1e (and repeated in Fig. R1 (a)).

To the contrary, as we have shown in the pulse length progression of Supplementary Figure S8, the non-linear signatures clearly survive when the VIS pulse is added.

The reason for this is simple: if the H₂⁺ wave packet is created before IR pulse most of the dissociation will occur on the rising edge of the pulse when the intensity is low. Thus, almost all of the dissociation will pass through the linear/conical/1-photon intersection, and the high-order intersection will not have a significant effect on the proton spectra.

In terms of accessibility of the high-order intersections the separated pulse limit corresponds to the experiments performed on with H₂⁺ beams from ion sources such as reported by Natan et al.² (ref. [23] in the manuscript).

Referee 3: “As I mentioned in my previous report, the observations presented here is mainly due to the modulation of the laser waveform of the temporally overlapped IR and VIS pulses, rather than the high order effect of the crossing potential energy surfaces induced by the IR pulse which does not require any temporal overlap of the VIS pulse.”

² A. Natan, M. R. Ware, V. S. Prabhudesai, U. Lev, B. D. Bruner, O. Heber, and P. H. Bucksbaum, Phys. Rev. Lett. **116**, 143004 (2016).

We have discussed above that (a) there is no dichotomy between non-linear transitions and multi-color laser fields, and (b) the separated laser pulses do not produce clear signatures of the intersections of non-linear light-induced potentials.

REVIEWERS' COMMENTS

Reviewer #3 (Remarks to the Author):

I have read the revised manuscript carefully, but did not find noticeable improvements as compared to the previous version. The present results, in particular the main result of figure 2(b), by using two temporally overlapped VIS and mid-IR pulses, cannot support their conclusion of the observation of high-order crossing intersections, e.g. figures 1(c-e) where only a single mid-IR pulse is required. The using of two temporally overlapped orthogonal two-color pulses, in particular when the intensity of the VIS is about 10 times of that of the mid-IR, the observed off-axis emission of the protons is dominantly due to the coherent absorption of different numbers of the photons from two colors. This effect will surely hide the physics of the high-order crossing intersections induced by the mid-IR pulse. The intense VIS pulse not only introduces H_2^+ by singly ionizing H_2 , but also leads to the emission of protons from the dissociation of H_2^+ working together with the perpendicularly polarized mid-IR pulse. It agrees with their result presented in figure 2(c) where no off-axis emission of proton is observed when parallel polarized VIS and mid-IR pulses are used, indicating the importance of the modulated waveform of the perpendicularly polarized VIS and mid-IR pulses.

I would like to repeat that, if the authors can observe similar results for the case that the probe IR pulse is delayed with respect to the intense VIS pump pulse without any temporal overlap, I will believe that they do observe the physical process they initially proposed and I will be happy to recommend the publication.

Reviewer #4 (Remarks to the Author):

The manuscript is prepared well and the authors report interesting experimental results with the theoretical interpretation based on the Floquet picture. The authors responded appropriately to the criticisms raised by the reviewer #3. I think the revised MS can be accepted for publication.

In the following we will address the report by Referee #3.

Referee 3: "I have read the revised manuscript carefully, but did not find noticeable improvements as compared to the previous version. The present results, in particular the main result of figure 2(b), by using two temporally overlapped VIS and mid-IR pulses, cannot support their conclusion of the observation of high-order crossing intersections, e.g. figures 1(c-e) where only a single mid-IR pulse is required. The using of two temporally overlapped orthogonal two-color pulses, in particular when the intensity of the VIS is about 10 times of that of the mid-IR, the observed off-axis emission of the protons is dominantly due to the coherent absorption of different numbers of the photons from two colors. This effect will surely hide the physics of the high-order crossing intersections induced by the mid-IR pulse. The intense VIS pulse not only introduces H_2^+ by singly ionizing H_2 , but also leads to the emission of protons from the dissociation of H_2^+ working together with the perpendicularly polarized mid-IR pulse. It agrees with their result presented in figure 2(c) where no off-axis emission of proton is observed when parallel polarized VIS and mid-IR pulses are used, indicating the importance of the modulated waveform of the perpendicularly polarized VIS and mid-IR pulses.

I would like to repeat that, if the authors can observe similar results for the case that the probe IR pulse is delayed with respect to the intense VIS pump pulse without any temporal overlap, I will believe that they do observe the physical process they initially proposed and I will be happy to recommend the publication.

Unfortunately, but not wholly unexpectedly, the referee once more chooses to ignore our answer to their previous criticism. We do not repeat here the arguments brought forward in our previous revisions, where the above points have been exceedingly discussed.

We have made a small change to the text on page 9, where we replaced

"For the visible few-cycle pulse, we also consider a weak pulse pedestal at 5% of the peak intensity and with a duration of 45 fs (full width at half-maximum of the intensity envelope)."

With

"Due to the observed importance of the visible field in shaping the experimental results, we also consider a weak pulse pedestal at 5% of the peak intensity and 45 fs duration (full width at half-maximum of the intensity envelope) for the visible few-cycle pulse."

In the following we will address the report by Referee #4.

Referee 4: "The manuscript is prepared well and the authors report interesting experimental results with the theoretical interpretation based on the Floquet picture. The authors responded appropriately to the criticisms raised by the reviewer #3. I think the revised MS can be accepted for publication."

We would like to thank the referee for taking the time to review our manuscript and the correspondence with referee #3.